# Modelling the thermal evolution of extensional basins through lithosphere stretching factors: application to the NW part of the Pannonian basin

Eszter Békési[1], Jan-Diederik van Wees[2], Kristóf Porkoláb[1], Mátyás Hencz[1], Márta Berkesi[1]

[1]MTA-EPSS Lendület (Momentum) FluidsByDepth Research group, HUN-REN Institute of Earth Physics and Space Science, 9400 Sopron, Hungary
[2]TNO Utrecht, Utrecht 3584 CB, Netherlands

*Correspondence to*: Eszter Békési (bekesi.eszter@epss.hun-ren.hu)

**Abstract.** The reconstruction of thermal evolution in sedimentary basins is a key input for constraining geodynamic processes and geo-energy resource potential. We present a methodology to reproduce the most important transient thermal footprints accompanying basin formation: lithosphere extension and sedimentation. The forward model solving the transient heat equation is extended with an inversion workflow to constrain models with temperature measurement, providing estimates on model parameters, most importantly the amount of lithosphere stretching. We apply the methodology to the NW part of Hungary. We test the effect of variations in model input parameters on the resulting temperature estimates and discuss the uncertainties and limitation of the modelling technique. Realistic past- and present-day temperature predictions for the entire lithosphere are achieved for a carefully assessed set of input parameters, suggesting the strong attenuation of the mantle lithosphere through extension, and relatively small variations in the present-day thermal lithosphere thickness. The new temperature model allows an improved estimation of lithosphere rheology and the interpretation of mantle xenolith origins.

## 1 Introduction

Understanding the thermal state and thermal evolution of the lithosphere of sedimentary basins are crucial both for constraining fundamental geodynamic, geological, and geochemical processes and observations on lithosphere scale, as well as for geo-energy perspectives such as geothermal and hydrocarbon exploration and resource characterization (e.g. Cloetingh et al., 2010; Ranalli and Rybach, 2005). Extensional sedimentary basins, through their formation, exhibit a typical thermal evolution pattern. During the active rifting phase, surface heat flow, lithosphere temperature and geothermal gradient rise, governed by the thinning of the lithosphere and consequent rise of the asthenosphere (e.g. Buck et al., 1988; Royden and Keen, 1980). Subsequently, the thermal relaxation of the lithosphere begins through conductive cooling and thermal subsidence. The duration of both the syn- and post rift phases vary significantly, however, reaching equilibrium (steady-state) typically takes several tens to hundreds of million years (Van Wees et al., 2009; Xie and Heller, 2009; Petersen et al., 2015).

In this paper we present a new methodology that accounts for the most important thermal effects that accompany basin formation such as lithosphere extension, sedimentation/erosion, and changes in thermal properties, most importantly the radiogenic heat generation in the upper crust, largely building on the methodology of Van Wees et al. (2009). The transient

thermal modelling workflow is extended with an inversion framework to constrain model parameters with present-day temperature observations, that allows the validation of the resulting model predictions. We demonstrate and apply the new methodology to the NW part of the Pannonian basin (Fig. 1).

The Pannonian basin exhibits an attenuated crust and lithosphere (Hetényi and Bus, 2007; Kalmár et al., 2021; Kalmár et al., 2023) and therefore high heat flow (an average of 90 mW/m$^2$) and geothermal gradient (an average of 45 ºC/km), constituting one of the hottest basins in Europe (Lenkey et al., 2002; Békési et al., 2018; Horváth et al., 2015; Limberger et al., 2018), together with the Tyrrhenian and Aegean basins (e.g. Mendrinos et al., 2010; Giovanni et al., 2005). Lithosphere extension in the Pannonian basin took place in the Miocene migrating from NW towards SE. Consequently, surface heat flow and

geothermal gradient in the NW part of the basin constituting the study area is generally lower, but the thermal footprint of extension is still notable. Extension was followed by post-rift cooling and subsidence accompanied by contractional basin inversion from the Late Miocene (e.g. Balázs et al., 2016; Fodor et al., 2005; Horváth and Cloetingh, 1996; Tari, 1994; Tari et al., 2020) to present day (Grenerczy et al., 2005; Bada et al., 2007; Porkoláb et al., 2023; Békési et al., 2023). Despite the inversional overprint, the thermal footprint of Miocene lithosphere extension is still the most important factor that determines

the present-day thermal state of the lithosphere. Consequently, the past and present-day temperature distribution in the lithosphere can only be fully captured by modelling the transient thermal effect of syn-rift extension and post-rift cooling, accompanied by changes in lithosphere structure and thermal properties (i.e. changes in thermal properties due to sedimentation, upper crustal radiogenic heat generation).

Physics-based thermal models constructed for (parts of) the Pannonian basin partly focused on the representation of the

temperature distribution within the upper crust, providing boundary conditions for geothermal exploration (Lenkey et al., 2017; Békési et al., 2018). Such models were constructed either without performing actual transient calculations (Békési et al., 2018) or were not conditioned by temperature measurements (only the forward modelling exercise was performed (Lenkey et al., 2017)). The thermal evolution of the lithosphere of (parts) of the Pannonian basin was also modelled (Balázs et al., 2021; Majcin et al., 2015), without the direct incorporation of temperature measurements. We aim to provide temperature predictions

that can further improve on existing models to represent past and present-day temperature distribution within the whole lithosphere. Additionally, we test the effect of a range of initial model parameters on the resulting thermal field and estimate the amount of lithosphere stretching in the area for a selected case of model parameters. We discuss implications for the thermal evolution of the region, as well as for the rheology of the lithosphere. Moreover, we outline its applications for geochemical measurements on mantle xenoliths.

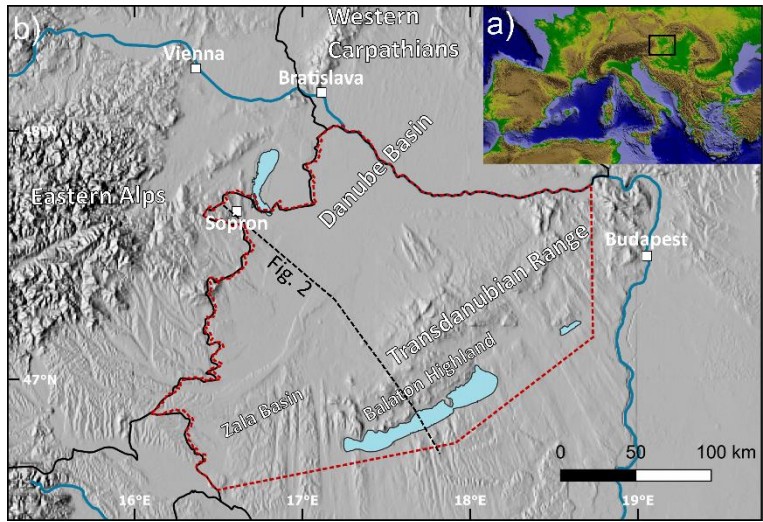


**Figure 1: a) Topographic map of Europe based on the SRTM digital elevation model (Farr et al., 2007) , showing the outline of Fig. 1b (black rectangle) b) Geography of the study area based on the GMTED2010 elevation model (Danielson and Gesch, 2011). Red polygon denotes the extent of the thermal model (restricted to the borders of Hungary due to data availability), black lines denote state borders.**

## 2 Geological setting


Our study area in NW-Hungary comprises sub-basins of the Miocene Pannonian basin system (Danube basin, Zala basin) and the Transdanubian Range, where the pre-Cenozoic basement units outcrop over a hilly region (Figs. 1, 2). The Danube basin (also called Little Hungarian Plain) is one of the deepest (up to 9 km (Kilényi and Šefara, 1989)) sub-basins of the Pannonian basin and is framed by the Eastern Alps to the west, the Western Carpathians to the north, and the Transdanubian Range to the

southeast. The sedimentary succession of the Danube basin overlies an Alpine nappe stack of basement units consisting of Adria-derived thrust sheets (Austroalpine nappe system), remnants of the Alpine Tethys ocean (Penninic nappe), and units of the lower plate (Europe-derived units). During the Miocene opening of the Danube basin, normal faults partially reactivated and partially cut through the Alpine nappe contacts in the basement (Tari et al., 2021). The Alpine nappe stack is exposed on the NW and SE margins of the Danube basin: the Lower Austroalpine nappe in the Sopron Mountains, while the Upper

Austroalpine units in the Transdanubian Range (Fig. 2 (Tari, 1994; Schmid et al., 2008)). The Transdanubian Range exhibits a thick Mesozoic platform carbonate succession (Fig. 2) that defines its characteristic thermal properties (Table 1) and typical karstic hydrology (Mádl-Szőnyi and Tóth, 2015). The SE limit of the Transdanubian Range is the Mid-Hungarian Shear Zone (Csontos and Nagymarosy, 1998), where basement units are buried below Neogene sediments (Fig. 2).

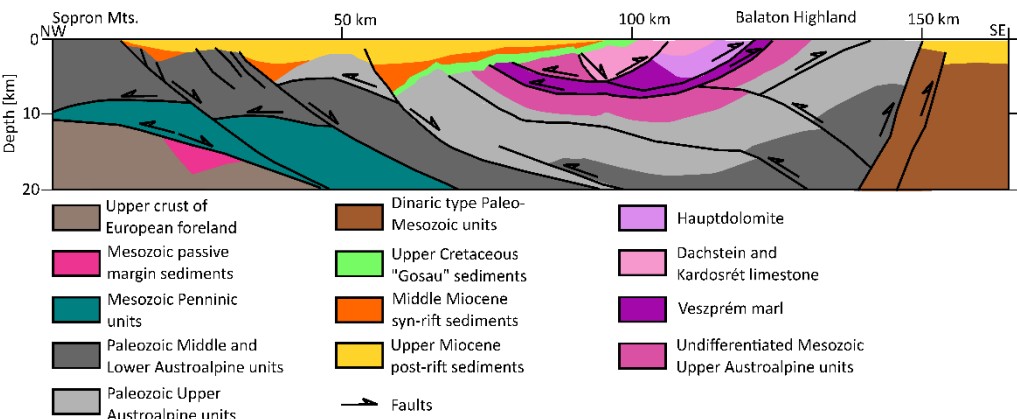

**Figure 2: Geological cross section through the study area (for location see Fig. 1) showing the most important regional units and faults, modified after Szafián et al. (1999).**

## 3 Data and methods

### 3.1 Model geometry and thermal properties

The temperature model extends to the whole lithosphere in the NW part of the Pannonian basin, restricted to its Hungarian part. Restricting the model area to the Hungarian part was necessary due to the availability of geological horizons and temperature measurements. The model was built in the Hungarian coordinate system (HD72 / EOV) with a horizontal resolution of ~ 3 km and a vertical resolution of 200 m for the uppermost 5 km and 2.5 km down to 135 km depth, which was selected as the bottom of the lithosphere prior to extension.

The model is built up by the present configuration of sedimentary layers, upper crust, lower crust, and lithospheric mantle. The sediments were subdivided into four layers; Quaternary, Upper Pannonian (Upper Miocene post-rift), Lower Pannonian (Upper Miocene post-rift), and pre-Pannonian Neogene (Middle Miocene syn-rift) and Paleogene units built up by the mixture of clastic sediments (Table 1). Paleogene sediments were not sub-divided from the pre-Pannonian Neogene sediments because of their limited overall extent in the study area, but was accounted for in the selection of the composition, based on Babinszki et al. (2023b). The compositions of sedimentary layers were determined based on interpreted seismic sections and well logs as well as derived geological models (Babinszki et al., 2023b; Fodor et al., 2013; Sztanó et al., 2016). For geometry of the pre-Cenozoic basement, we followed Haas et al. (2014). We included an additional layer for the Mesozoic carbonate basement units, since they constitute relatively thick (up to a few kms) successions throughout parts of the study area and have significantly different thermal properties compared to crystalline basement units. We constructed a thickness map (https://data.mendeley.com/drafts/vp7jdp79y4) and a composition ratio for the carbonates based on published cross-sections and geological models (Budai et al., 1999; Szafián et al., 1999; Héja et al., 2022; Haas et al., 2014; Babinszki et al., 2023a). For the depth of the lower and upper crust in the present-day model, we used the most recent crustal models constructed from seismological observations (Kalmár et al., 2021). Except for the starting model for the time dependent calculations

(representing the thermal state of the lithosphere prior to extension), we allowed the lithospheric mantle to stretch with a spatially variable factor (subcrustal stretching factor, see section 3.3) instead of using any present-day lithospheric thickness

maps. We tested a range of initial lithosphere and crustal thickness values (Table A1) to evaluate the effect of initial parameter selection on the resulting modelled temperatures. The depth of the lithosphere-asthenosphere boundary (LAB) prior to stretching, corresponding to 1330 ºC, was set to a constant 135 km, and the initial crustal thickness was set to 40 km for the preferred model. These initial conditions are considered suitable to represent the overthickened pre-extension lithosphere of the Alpine-Carpathian region (e.g. Faccenna et al., 2014). The initial crustal thickness of 40 km was chosen as an input value

that is realistic for most of the study area, however, crustal thickness was possibly even larger in the western periphery of the study area, most importantly in the area of the Rechnitz core complex.

For the calculation of thermal conductivities of the sediments, we used matrix thermal conductivity values for shale and sandstone (pelite and psammite) in the Pannonian basin (Dövényi and Horváth, 1988), and typical values after Hantschel and Kauerauf (2009) for conglomerate and marl. The matrix thermal conductivities were corrected for in-situ temperature using

the formula of Sekiguchi (1984). For the carbonate layers built up dominantly by dolomites and limestones (Table 1), we adopted values reported in Dövényi et al. (1983). Since each sedimentary layer and the carbonate layer are built up by various lithotypes, the bulk rock matrix thermal conductivities were calculated by taking the harmonic mean of the individual matrix thermal conductivities of the lithotypes. The sediment bulk thermal conductivities were finally obtained using the geometric mean of the bulk matrix conductivities and the thermal conductivity of the pore fluid as described e.g. in Limberger et al.

(2018) . For the calculation of porosity of sediments, we estimated compaction coefficients and depositional porosities based on the porosity-depth trends of Szalay (1982) for shale and sand(stone) (pelite and psammite), and adopted typical values reported by Hantschel and Kauerauf (2009) for conglomerate and marl.

The ranges of thermal conductivity values of Neogene sediments vary between 1.4 and 2.4 (Table 1), which is lower than the mean measured thermal conductivity values of shale and sandstone samples reported in Mihályka et al. (2023). This can partly

be explained by the low thermal conductivity of highly porous unconsolidated quaternary and Upper Pannonian sediments in shallow depth, as well as the dominance of shales with low thermal conductivity in the Lower Pannonian layer.

We calculated the thermal conductivities of the crust and the lithosphere using the thermal and petrophysical parameters of Hantschel and Kauerauf (2009). Typical thermal conductivity values of the upper and lower crust and lithospheric mantle were corrected for pressure- and temperature conditions based on Chapman (1986) in case of the crust, and Schatz and Simmons

(1972) and Xu et al. (2004) for the mantle lithosphere.

Radiogenic heat generation of each layer was calculated as a mixture of typical values of lithotypes after Hantschel and Kauerauf (2009), corrected for compaction (values in sediments generally increase with depth due to decreasing porosity). The radiogenic heat generated in the granitic upper crust is generally considerably larger than in case of sedimentary, lower crustal and lithospheric mantle units. Therefore, it was increasingly important to distinguish the carbonate and crystalline basement

units for the proper prediction of upper crustal temperatures. Since the radiogenic heat generation of compacted shale layers is in the order of magnitude of the upper crust, maximum values of the sediment heat generation corresponding to the deep

Lower Pannonian shales is up to 1.7 µW/m$^3$ (Table 1). The radiogenic heat generation of the crust and lithospheric mantle were selected to constants. For the upper crust, we chose a typical continental upper crustal heat generation value of 1.4 µW/m$^3$, while the lower crustal and mantle lithosphere heat generation was selected to 0.4 µW/m$^3$ and 0.002 µW/m$^3$ based on Hantschel and Kauerauf (2009) (Table 1).


| Layer name | Lithology | Thermal conductivity [W/m*K] | Radiogenic heat production [µW/m$^3$] |
|---|---|---|---|
| Quaternary | 70% sand; 30% shale | Bulk values per lithotypes (mixed lithologies) based on Hantschel and Kauerauf (2009) and Dövényi and Horváth (1988), dependent on compaction and temperature; ranging between 1.4-2.4 | Bulk values per lithotypes based on Hantschel and Kauerauf (2009) dependent on compaction; ranging between 0.4-1.7 |
| Upper Pannonian (Upper Miocene) | 50% sand; 50% shale | | |
| Lower Pannonian (Upper Miocene) | 30% sand; 70% shale | | |
| Neogene and Paleogene (pre-Pannonian) | 35% sand; 35% conglomerate, 15% limestone, 15% marl | | |
| Mesozoic carbonate | 30% limestone; 60% dolomite; 10% sand | Bulk values per lithotypes (mixed lithologies) based on Hantschel and Kauerauf (2009) dependent on compaction and temperature; ranging between 2.7-3 | Bulk values per lithotypes based on Hantschel and Kauerauf (2009) dependent on compaction; ranging between 0.3-0.4 |
| Upper crust | 100% granite | Bulk values per lithotypes (Hantschel and Kauerauf, 2009) corrected for pressure and temperature (Chapman, 1986); ranging between 2-2.8 | Constant based on Hantschel and Kauerauf (2009); 1.4 |
| Lower crust | 100% granulite | | Constant based on Hantschel and Kauerauf (2009); 0.5 |

| Mantle lithosphere | 100% peridotite | Bulk values per lithotypes (Hantschel and Kauerauf, 2009) corrected for pressure and temperature (Schatz and Simmons, 1972; Xu et al., 2004), ranging between 2.8-3.5 | Constant based on Hantschel and Kauerauf (2009); 0.02 |
|---|---|---|---|

**Table 1: Lithology and thermal properties of model layers.**

### 3.2 Temperature observations and data uncertainties

We calibrated the thermal model with subsurface temperature measurements from hydrocarbon and geothermal wells.
Measurements from the Geothermal Database of Hungary (Dövényi and Horváth, 1988; Dövényi et al., 2002) the Geothermal Information System (Ogre, 2020) were collected, including bottom hole temperatures (BHTs), drill-stem tests (DSTs), steady-state temperature logs and outflowing water temperatures from geothermal wells. Temperature measurements were carefully reviewed and observations from areas where the conductive thermal field is strongly influenced by fluid flow and observations with errors larger than 10 ºC were excluded from the dataset. This was necessary as the model, focusing primarily on
lithosphere-scale processes, could not account for convective heat transfer, and temperature measurements influenced by fluid flow would have biased the predicted lithosphere temperatures. The influence of fluid flow was checked on the individual temperature measurements of wells as well as on the shallow (500 m) temperature map (Lenkey et al., 2021). The resulting number of temperature observations used for calibration was 319, covering the depth interval of 200-5100 meters (Fig. 3a, https://data.mendeley.com/drafts/vp7jdp79y4). Measurements are not evenly distributed throughout the study area; most of
them are available from basinal locations, especially from the surroundings of the Zala basin (Figure 3b). Observations from the vicinity of the Transdanubian range are rather limited due to the presence of regional deep fluid pathways (Mádl-Szőnyi and Tóth, 2015; Tóth et al., 2023) and resulting convective thermal field, also evidenced by the low surface heat flow due to the infiltration of cold meteoric water (Lenkey et al., 2002).

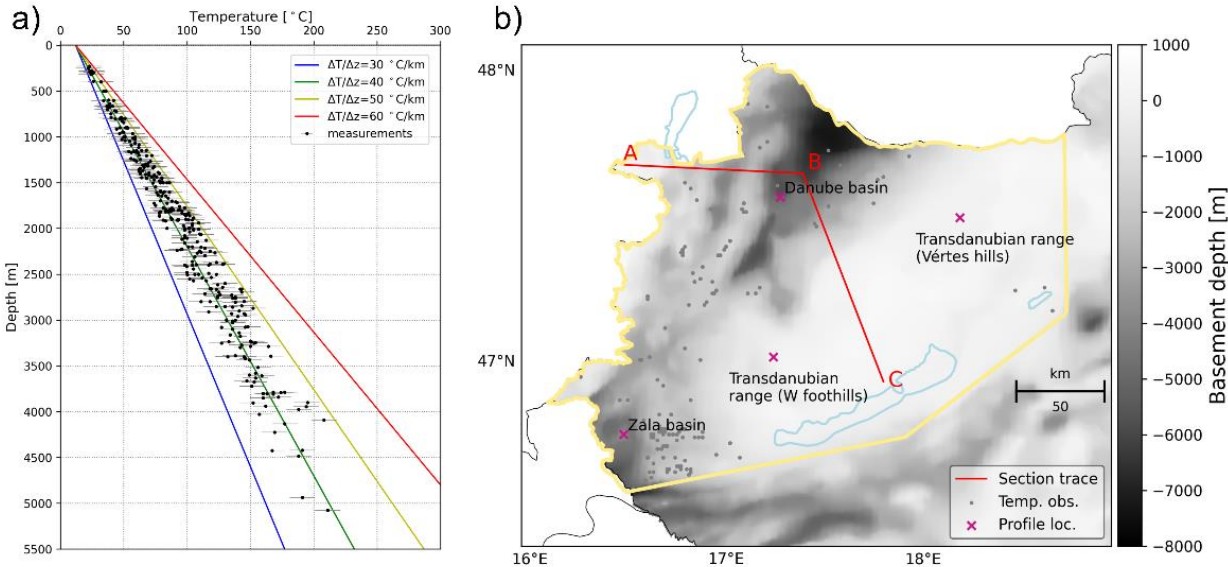

Figure 3: (a) Temperature dataset used for the calibration of the thermal model. Temperature measurements were obtained from the Geothermal Database of Hungary (Dövényi and Horváth, 1988; Dövényi et al., 2002) and the Geothermal Information System (Ogre, 2020). Colours represent geotherms between 30 ºC/km to 60 ºC/km. (b) Locations of temperature measurements (grey circles), and locations of temperature profiles (pink crosses) and section (red line) shown in Figs. 7-9, plotted on top of the pre-Cenozoic basement map (Haas et al., 2014).

Symmetrical uncertainties were chosen for the measurements, between ±5 to ±10 ºC, and uncertainties were selected identical for the same measurement types for simplicity, similar to previously published studies (Békési et al., 2018; Békési et al., 2020). DSTs were marked by uncertainties of ±5 ºC, while for BHTs and outflow temperatures, a maximum error of ±10 ºC was chosen. For the remaining temperature measurements, we adopted the errors reported in the Geothermal Database of Hungary (Dövényi and Horváth, 1988; Dövényi et al., 2002).

Temperature measurements selected for calibration mostly scatter around the 40 ºC/km geotherm (Fig. 3a), while several observations, both in shallower and deeper intervals, approximate the 50 ºC/km geotherm. The overall geothermal gradient of the temperature dataset is 42 ºC/km, which is slightly below the average geothermal gradient for the central part of the Pannonian basin (~45 ºC/km), although still much higher than the average continental values, representing the thermal effect of the thinned lithosphere in the study area.

### 3.3 Forward model

The modelling procedure consists of three main steps, including steady-state conductive forward model calculations, transient calculations incorporating the thermal effect of lithosphere-scale processes, and the inversion procedure. In the first step, we calculated the thermal field prior to lithosphere extension (Section 3.3.1). In the second step, we used crustal and subcrustal stretching factors and sedimentation rates to account for the effects of lithosphere extension and subsequent cooling, as well as syn- and post-rift sedimentation (Section 3.3.2) damping of the thermal footprint of extension. The third step concerns the

inversion workflow (Section 3.4), incorporating temperature measurements into the model as target observations to constrain the amount of lithosphere attenuation and as a result obtain more realistic temperature estimates during and after rifting.

### 3.3.1 Steady-state calculations

The steady-state modelling approach provides initial conditions for the transient model calculations, by solving the heat equation for conduction in 3D:

$$0 = \nabla \cdot (\lambda \nabla T) + A \tag{1}$$

where $\lambda$ is the thermal conductivity [Wm$^{-1}$ K$^{-1}$], $T$ [K or °C] is the temperature, $A$ is the radiogenic heat production [Wm$^{-3}$], and $\nabla = \left(\frac{\partial}{\partial x}, \frac{\partial}{\partial y}, \frac{\partial}{\partial z}\right)$ is the nabla operator. Equation (1) is solved numerically by a finite-difference approximation using the Preconditioned Conjugate Gradient method. Temperature boundary conditions on the top and bottom of the model were selected as 12 °C and 1330 °C, respectively. The top boundary condition of 12 °C was selected as a mean surface temperature. The depth of the bottom boundary condition was selected to 135 km, which was assumed to be the depth of the LAB prior to lithosphere extension. The vertical edges of the model were assumed to be insulating with a fixed heat flow of zero. These boundary conditions remained active also for the transient model calculations both with and without incorporating the inversion procedure, since the steady-state model provided the initial setting of the transient modelling. Please note that the steady state geotherm is based on the present day (actual) crustal and sediment configuration in target prediction time (present day). As demonstrated in Van Wees et al. (2009) in high resolution 1D simulations, the steady state solution at prediction time target, corrected for transient effects related to kinematic effects of lithosphere deformation, and sedimentation provide a reliable thermal solution for in particular in the top 5-10 km of the model.

### 3.3.2 Transient calculations

To correct the steady state solution (Equation 1) for transient effects, the thermal effects of lithosphere extension was incorporated in the model by integrating over simulation time for:

$$\frac{\partial T}{\partial t} = 1/\rho c_t \cdot [\nabla \cdot (\lambda \nabla T) + A] - v_z \partial T / \partial z \tag{2}$$

where $t$ is time [s], $\rho$ is density [kgm$^{-3}$], $c_t$ is specific heat capacity [J kg$^{-1}$ K$^{-1}$], $v_z$ is vertical velocity of the sediment, crust and mantle in the Eulerian finite difference framework as a function of the tectonic stretching and sedimentation (cf. Van Wees et al., 2009; Bonté et al., 2012; Corver et al., 2009). The transient term was estimated based on crustal ($\delta$) and subcrustal ($\beta$) stretching factors and accounting for sedimentation, based on Van Wees et al. (2009). Crustal and subcrustal stretching factors represent the ratio between the initial and thinned crustal thickness and mantle lithosphere thickness, respectively, with values >1 (e.g. Royden and Keen, 1980). For the transient numerical modelling of the temperature evolution of equation (2), a 3D

explicit 3-step Runge-Kutta finite difference approach was used (Verwer, 1996) with a finite volume approximation. For instance, in case of an initial crustal thickness of 30 km, and a thinned crustal thickness of 20 km, δ equals to 1.5 (Fig. 4).

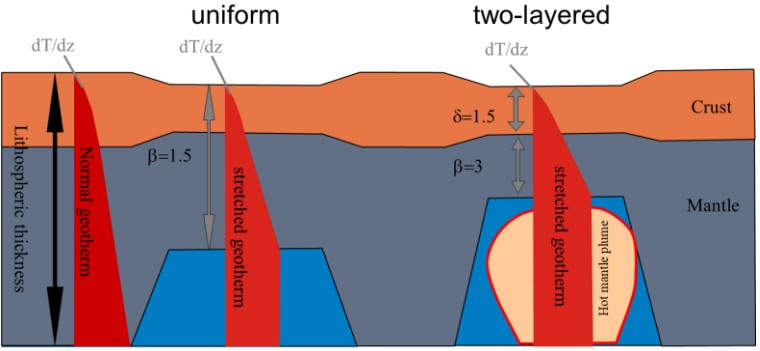

**Figure 4: Cartoon illustrating the crustal (δ) and subcrustal (β) stretching factors. dT/dz represents the temperature gradient with depth showing a disturbed geotherm in the stretched part of the model. Non-uniform stretching of the crust and mantle lithosphere (with or without the presence of mantle plumes) can be accounted for by β > δ after Van Wees et al. (2009) and Corver et al. (2009).**

The timing of the main extensional phase was not uniform in the study area. Highest rates in the Zala basin are inferred between 19-15 Ma, while in western part of the Transdanubian Range active normal faulting started only at ~15 Ma and persisted until 8 Ma (Fodor et al., 2021). In the Danube basin, the syn-rift phase was active between ~16-10 Ma (Šujan et al., 2021). In the thermal model, we assumed a uniform timing for active rifting in the whole study area for simplicity, which took place between 18-10 Ma (Table 2). It was necessary to later invert for subcrustal stretching factors in one step. For this period, we also considered sedimentation corresponding to the deposition of Pre-Pannonian Neogene sediments (Table 2).

| Tima [Ma] | Initial crustal thickness [km] | Initial LAB depth [km] | Crustal stretching (δ) [-] | Subcrustal stretching (β) [-] | Sedimentation [km] |
|---|---|---|---|---|---|
| 18 — 10 | 40 | 135 | Spatially variable calculated from the initial and present-day Moho depth and basement depth, ranging between ~1.2 to 2.6 | Constant value of 4 | Neogene (pre-Pannonian) sediment thickness, ranging between ~0-5 |
| 10 — 0 | - | - | 1 | 1 | Pannonian and Quaternary sediment thickness, ranging between ~0-5 |

**Table 2: Input parameters of the stretching module.**

During the active rifting phase, we calculated the transient thermal effect of extension using crustal (δ) and subcrustal (β) stretching factors for the area. Lenkey (1999) calculated these factors for the entire Pannonian basin, although after testing them we decided not to use them, due to the low β values predicted for the Transdanubian Range, resulting in unrealistically low present-day temperatures (almost identical with the thermal field prior to extension) in the area. We calculated new crustal stretching values similar to the methodology without heat flow observations described in Lenkey (1999) but based on the most recent present-day Moho depth of Kalmár et al. (2021) ($z_{Moho\ present}$). To be able to compare the new δ grid with the earlier work of Lenkey (1999), we chose an initial crustal thickness ($z_{crust\ init}$) of 40 km. We calculated the present-day crustal thickness using the present-day basement depth (Haas et al., 2014). The equation for the crustal stretching factor δ is the following:

$$\delta = \frac{(z_{crust\ init} - z_{basement})}{z_{Moho\ present}})$$  (3)

The resulting crustal stretching factors are between ~1.2 to 2.6 (Fig. 5a), where smaller values indicate almost no thinning of the crust corresponding to areas with no or minor sediment coverage, while highest values are attributed to basinal locations. Subcrustal stretching values cannot be calculated in the same way as the crustal stretching but using the present-day LAB depth, since the base of the lithosphere immediately after extension has considerably changed through post-rift cooling (Lenkey, 1999). Therefore, we selected constant prior values for β, which we updated through the inversion procedure (Section 3.4) to account for its potential spatial variations. We tested several starting values for β between 2 and 4 (Appendix A), and finally we chose β=4, since this value provided the prior model best fitting to temperature observations. In comparison with previous lithosphere thermal modelling studies for the Danube basin, for instance Majcin et al. (2015) used β value of 1 to 3, however with lower initial lithosphere thickness (120 km). Considering the initial lithosphere thickness of 135 km, and an initial crustal thickness of 40 km, β=4 would mean that the thickness of the mantle lithosphere reduced from 95 km to ~ 24 km during rifting. The active rifting phase was followed by post-rift thermal subsidence and corresponding post-rift sedimentation. We incorporated the effect of post-rift sedimentation by assuming constant sedimentation rates between 10 – 0 Ma, based on the thickness of Pannonian (Upper Miocene) and Quaternary sediments (Table 2). Post-rift cooling was incorporated in the model by defining stretching of 1 after the syn-rift period.

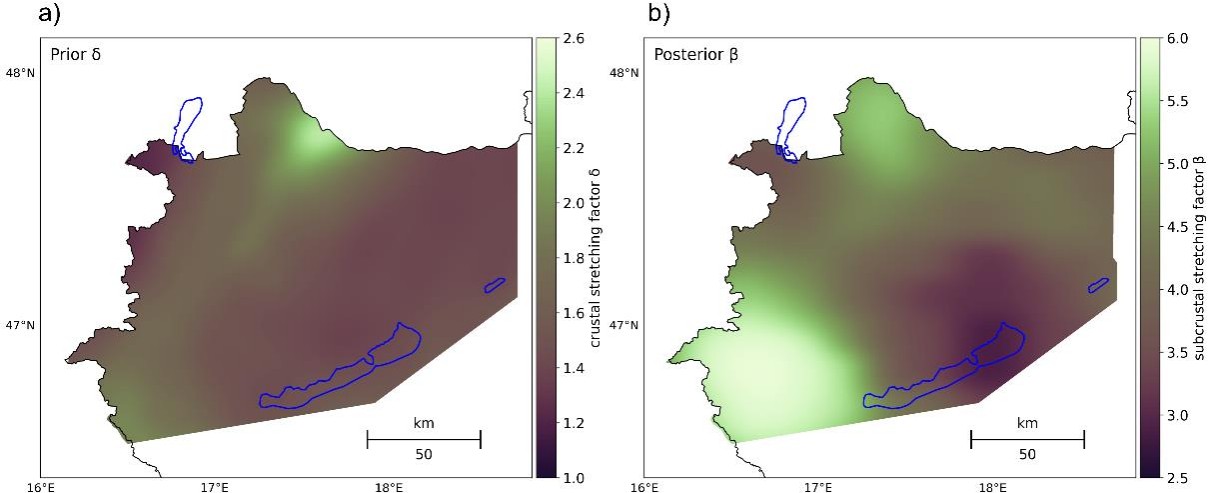

**Figure 5: (a) Prior crustal stretching (δ) and (b) posterior subcrustal stretching (β) values representing the extension of the crust and mantle lithosphere. Note that δ shown in (a) and β =3 were used as input parameters for the stretching module, and β shown in (b) is the posterior mantle stretching factor resulting from the inversion procedure, conditioned with temperature observations.**

### 3.4 Inversion procedure

We conditioned the thermal model with temperature observations from wells, using a selection of temperature measurements with assigned uncertainties described in Section 3.2. Since only one observation pre grid cell is supported, observations were restricted to 200 m deep intervals, and measurements with lower uncertainties were considered. I case the case of multiple observations with the same error per grid cell, the deeper one was used for calibration. During the inversion procedure, the only model parameter we updated was the subcrustal stretching factor β. We selected only β for the model update as we were primarily interested in lithosphere-scale thermal field and thermal evolution. We did not update the shallower part of the model (e.g. thermal parameters of the sediments) since an already good fit with temperature observations was achieved by only modifying β, that is responsible for the large-scale thermal perturbations affecting the model area.

To estimate the subcrustal stretching factor (β), we applied ensemble-based probabilistic inversion. The Ensemble Smoother (ES, Emerick and Reynolds, 2013a) estimates the model parameters by a global update, incorporating all data available. This allows for the solution of inverse problems with large number of observations in a computationally efficient way. For non-linear forward models, the ES requires several iterations, where the prediction of the previous run is used as an input for the subsequent data assimilation step (ES-MDA, Emerick and Reynolds, 2013b).

The solution for a single data assimilation for the updated model ensemble is:

$$\widehat{M} = M + M'[GM']^T \{GM'[GM']^T + (N_e - 1)C_d^{-1}\}^{-1} \times (D - GM) \tag{4}$$

In equation 4, $M$ is the prior ensemble of model parameters, $GM$ is the result of the forward model working on all ensemble members, and $GM'$ is the difference between $GM$ and its mean. $N_e$ represents the number of ensembles, and $D$ is an ensemble

of data realizations, created by perturbing the measurements according to their covariance matrix ($C_d$). The mean of the ensemble is taken as the best estimate, which is used as input for the next update in case of ES-MDA. The number of data assimilation steps, $N_a$ must be selected a-priori. The data covariances used for the update steps are increased by a multiplication factor, $\alpha_i$ for i=1,2…, $N_a$, and $\alpha_i$ must be selected as $\sum_{i=1}^{N_a} \frac{1}{\alpha_i} = 1$  (Emerick and Reynolds, 2013b). This is necessary to compensate for the effect of multiple applications of an ES.

The prior uncertainty in β was taken into account by scaling the initial β values of 3 to a uniform distribution between 2 and 6. The spatial variability of β was determined through a spherical variogram, representing the variability of subcrustal stretching as a function of distance. The radius of the variogram includes 15 model cells, which corresponds to ~45 km. This relatively large distance was selected because variations in subcrustal stretching were considered to be large-scale. During the ES-MDA procedure, we chose 4 iterations, each with 700 model runs (ensembles). The resulting β field (Fig. 4b) shows

variations between 2.5-6, where largest values correspond to the Zala basin, and the areas marked by less intense subcrustal stretching are predicted for the Transdanubian Range and the NW part of the model area.

## 4 Results

### 4.1 Shallow (0-5 km) temperature field

Present-day posterior model temperatures, calculated with the updated subcrustal stretching factors, β, are in general higher in

basinal areas (Zala basin, Danube basin) and lower in peripheral areas (Transdanubian Range, Sopron Mts.) (Fig. 6.). The largest positive thermal anomaly at 2 km depth corresponds to the Zala basin in the SW, reaching up to 100 ºC (Fig. 6, left panel). The pattern of anomalies at 4 km depth is slightly different: a pronounced positive anomaly also shows in the Danube basin in the north, with temperatures up to 170 ºC, meaning a geothermal gradient of ~39.5 ºC/km. Since convection connected to fluid flow is not considered in the model, the modelled thermal anomalies can be explained with conductive thermal effects.

Positive anomalies are the reflection of sediment blanketing, meaning the insulating effect of sediments in shallower depth, with low thermal conductivity. Negative anomalies can be attributed to outcropping/near-surface basement rocks (mostly carbonates) having significantly higher thermal conductivities, as well as lower lithospheric stretching relative to the basin areas (Fig. 5b). It is important to note that the conductive thermal modelling approach is a valid assumption for the majority of the study area, resulting in realistic predicted temperatures. The conductive assumption is although not fully valid for parts

of the Transdanubian range built up by fractured and karstified carbonate rocks, as well as in buried carbonates in the vicinity of the Transdanubian Range. Groundwater flow within the top 5 km alters the conductive regime at these areas, and therefore predicted temperatures cannot be considered reliable in the shallow part of the model. Misfits between modelled and observed

temperatures do not indicate this bias, since temperature measurements affected by fluid flow were excluded from the calibration dataset to properly account for the transient effect of lithosphere extension (see section 3.2).

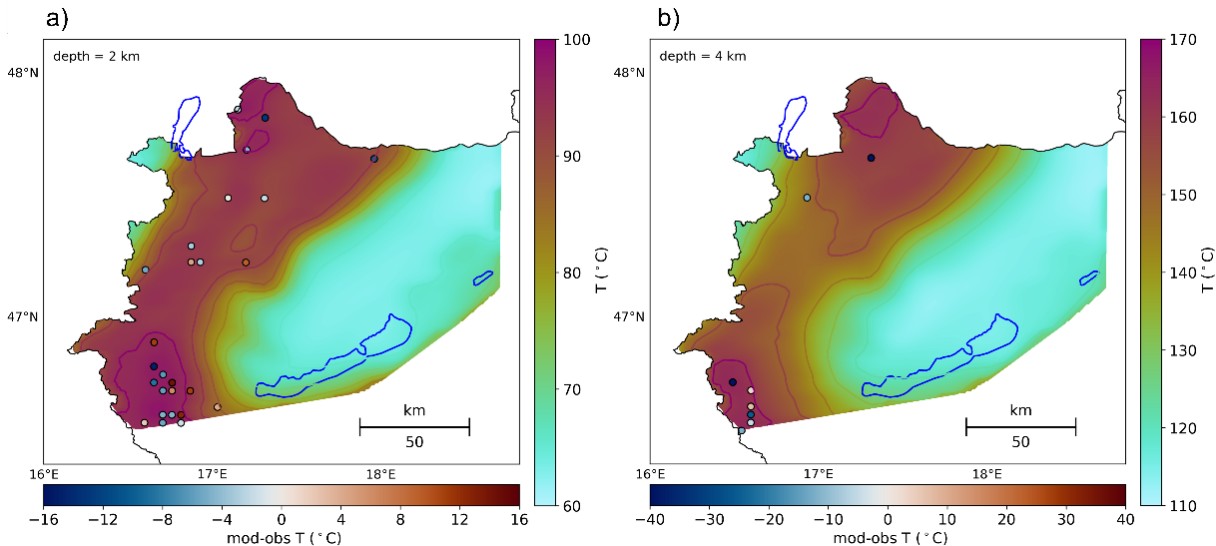

**Figure 6: (a) Isodepth temperature maps predicted by the present-day posterior model at 2 km (left panel) and 4 km (right panel) depth. The misfits between modelled and observed temperatures are indicated with color-coded circles, within the depth interval of ±200 m.**

The effect of sediment blanketing in shallow (0-5 km) depth is also clearly visible on the temperature-depth profiles (Fig. 7.). Temperatures are higher in basinal profiles (Fig. 7 a, b) than in marginal settings (Fig. 7 c, d), which is a result of the combined effect of sedimentation and higher crustal and lithospheric stretching in the basins. In all cases, the thermal effect of lithosphere extension is clearly visible: temperatures prior to stretching (Fig. 7, black lines) are significantly lower than present-day geotherms (Fig. 7, blue lines). Modelled present-day temperatures show a generally good fit with observations, although misfits in the deeper (>~3.5 km) exist in both the Danube and Zala basins. Some of these misfits may be explained by measurement errors but may also be attributed to changes in sediment geometry and composition further away from the profile location or can even be caused by local fluid convection e.g. in the carbonate basement (Fig. 7c).

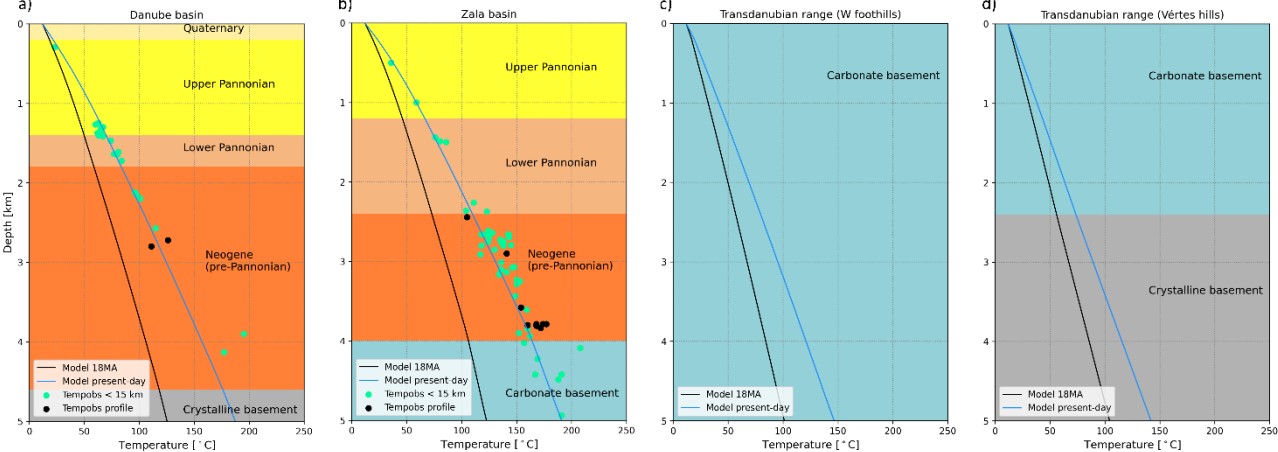

**Figure 7: Shallow (0-5 km) temperature-depth profiles in the Danube basin (a), Zala basin (b), and from two locations within the Transdanubian Range (c: western foothills, d: Vértes hills). Blue line represents the present-day geotherm, black line shows the geotherm prior to lithosphere extension. Black circles show temperature measurements from wells at the location of the profile, while green circles indicate measurements from wells within 15 km distance. For the locations of the profiles see Fig. 3.**

### 4.2. Lithosphere thermal field

The transient thermal field in the whole lithosphere was calculated by stretching the initial thermal model prior to extension (representing the thermal state of the lithosphere at 18 Ma) using crustal (δ) and subcrustal (β) stretching factors described in section 3.2. β was initially set to a constant value for the prior modelling, then a spatial variation of β was introduced and β values were updated to fit present-day model temperatures to temperature observations (described in detail in section 3.2). The resulting updated β values vary between 2.5 and 6 (Fig. 5.), suggesting that more than half of the initial mantle lithosphere was attenuated during extension in the entire area.

Lithosphere geotherms prior to stretching at 18 Ma (black lines in Fig. 8.) are significantly colder than past extension geotherms. The initial geotherms at 18 Ma indicate variations in geothermal gradient at two major compositional variations (sediment/basement and upper/lower crust boundary) according to the present-day model geometry. This is explained by the fact that present-day upper crustal geometries were used as a primary model input, since this setting provided the most appropriate initial conditions for the stretched models. Since no sediments and a thicker upper crust existed before extension, the initial thermal model representing the temperature field at 18 Ma is slightly biased in upper crustal levels. Going deeper, predicted initial lithosphere temperatures are almost identical for all locations (Fig. 8. a-d), that agrees with expectations that no major lateral temperature variations are expected in the lithosphere at 18 Ma.

We present the modelled thermal field affected by lithosphere extension for various representative time intervals (10 Ma, 8 Ma, 4.5 Ma, 2 Ma, 0 Ma, Fig. 7). All temperature profiles reach 1330 ºC at the depth of 120 km associated with the LAB,

prescribed as a bottom boundary condition for all models. The actual post stretching LAB is significantly shallower, as
suggested by the 10 – 0 Ma geotherms. Since heat transport processes are only considered in the lithosphere and not in the asthenosphere, the post-stretching models are only applicable in the thinned lithosphere. The present-day LAB (Kalmár et al., 2023) plotted on each profile therefore indicates the approximate depth until the models can be considered reliable (Fig 8.). Highest temperatures in the lithospheric mantle are attributed to the 10 Ma model (purple line in Fig. 8.), representing the thermal state right after extension. 10 – 0 Ma models represent the conductive cooling (thermal relaxation) of the lithosphere.
Cooling is combined with the thermal effect of post-rift sedimentation, that is most pronounced at the shallower parts of the models in basinal locations (Fig. 8 a, b). Present-day lithosphere temperature predictions as well as the elevated geothermal gradient and surface heat flow of the area (Lenkey et al., 2002) evidence that the thermal state of the lithosphere has not yet reached steady-state.

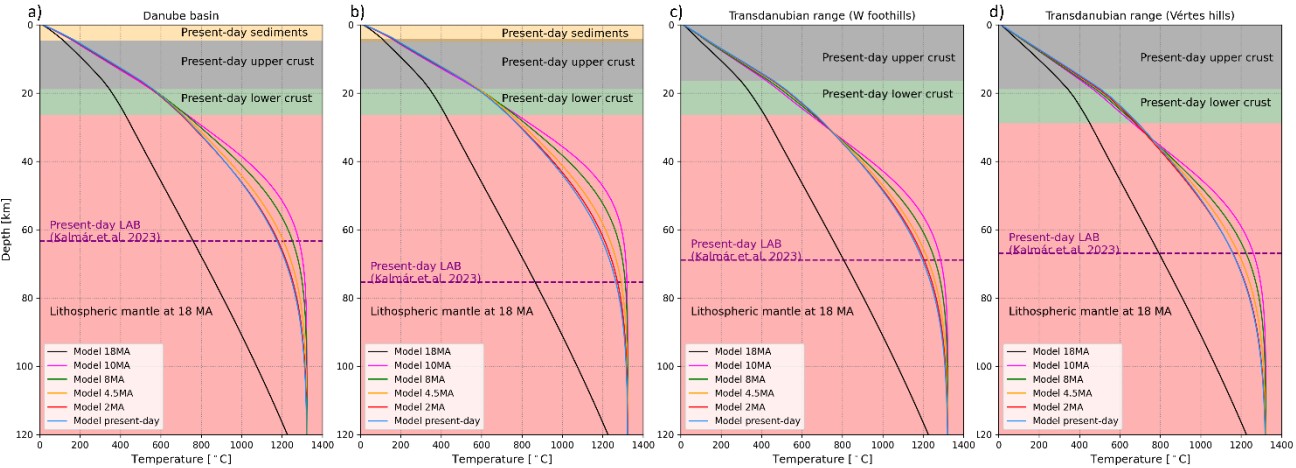

**Figure 8: Lithosphere temperature-depth profiles in the Danube basin (a), Zala basin (b), and from two locations within the Transdanubian Range (c: western foothills, d: Vértes hills). Color-coded lines represent geotherms from different times between 18 Ma – present. The depth extent of major units is also indicated, together with the present-day LAB (dashed purple line) from Kalmár et al. (2023), that is the approximate maximum depth where post-stretching models are considered reliable. For the locations of the profiles see Fig. 3.**

Present-day modelled temperatures are generally (slightly) elevated in basinal areas than the peripheral locations throughout the entire lithosphere (Figs. 8, 9). Higher temperatures in the Danube basin through the temperature profile in Fig. 9 represent the combined effect of lithosphere extension (controlling the thermal field in the mantle lithosphere) and sediment blanketing (having major influence in the crustal thermal field). Elevated deep lithosphere temperatures in the Danube basin can be explained by higher subcrustal thinning (Fig. 5b). Lithosphere temperatures reach 1200 ºC in the depth of around 70 km, which
agrees with the average LAB depth along the section (Kalmár et al., 2023).

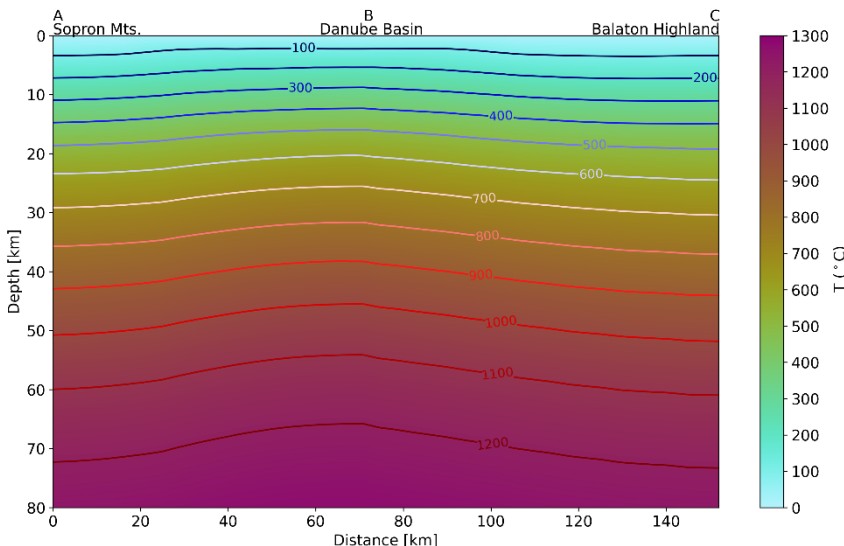

**Figure 9: Lithosphere temperature cross-section representing present-day predicted temperatures from the Sopron Mts. through the Danube basin to the Balaton Highland. For the location of the section see Fig. 3.**

**5 Discussion**

**5.1 Model uncertainties and limitations**

To quantify the added value of the inversion procedure through updating the subcrustal stretching factor (β), we compared the overall misfit between modelled and observed temperatures of the present day prior (β=4) and posterior model (inversion with spatial variation in β). Through the inversion, the median misfit has decreased from -3.13 ºC to -0.64 ºC. The RMS of the posterior model also decreased from 1.53 to 1.35, but this decrease is less significantly. More significant improvements of the misfit, especially in terms of the RMS where positive and negative errors do not cancel out, could be achieved by updating the thermal properties of the shallower part of the model (e.g. thermal conductivity of sediments, radiogenic heat generation in the upper crust). This exercise was excluded from the current study, as here we focus mainly on lithospheric scale thermal processes and thermal evolution of the lithosphere, which is primarily captured by the crustal and subcrustal stretching factors. We tested the influence of initial crustal and lithosphere thickness as well as the selection of the prior subcrustal stretching factors to modelled temperatures by performing a sensitivity analysis (Appendix A). This was necessary to select realistic input parameters. The initial crustal thickness of 35, 40, and 45 km, and initial lithospheric thickness of 120, 135 and 150 km were tested, with constant prior β value of 2, 3 and 4 (Table A1). All models showed the smallest RMS misfit with the highest tested β, while the influence of initial curtal and lithosphere thickness on resulting temperature predictions and associated RMS errors were less significant (Table A1, Figure A1). Therefore, β=4 was used as the final prior model presented through this

study (corresponding to Model 2c in the parameter test, Appendix A). The uncertainty of posterior β values resulting from the inversion procedure are estimated in terms of standard deviation (https://data.mendeley.com/drafts/vp7jdp79y4), with values up to 1.2. The standard deviation of beta values therefore provides a qualitative estimate of uncertainty in relationship of observed temperatures and subcrustal lithosphere model effects. This is important to note that standard deviations cannot fully capture the overall uncertainty of the estimated subcrustal stretching due to further model input parameter selections based on

assumptions, discussed in the following paragraph.

The prior and posterior models represent a specific case where several input parameters (initial conditions, thermal parameters, sedimentation rates, crustal stretching) were fixed. The selection of these parameters was performed carefully, while the uncertainties of these input assumptions cannot be neglected and therefore the resulting model predicts lithosphere temperatures that are specific to this case. Simplifications such as assuming a uniform timing of the extension for the entire

study area, application of constant sedimentation rate, and the indirect consideration of the basin inversion stage. Extension started and ceased significantly earlier in the Danube basin (Šujan et al., 2021) compared to the Zala basin and Transdanubian range (Fodor et al., 2021). The neglection of this difference in the timing of rifting could potentially result for instance in the underestimation of subcrustal stretching in the Danube basin. The assumption of constant sedimentation rate is valid for the basinal areas of the model, but are not fully valid for basin peripheries, where erosion due to basin inversion took place (e.g.

Szafián et al., 1999). This assumption can therefore result in more uncertain temperature predictions in the basin peripheries. Still, the geometry and structure of the uplifted basin margins were taken into account by the present-day crustal geometry that is used as a model input, which has the most important influence on the resulting thermal field. Further effects of the neotectonic inversion on the temperature field were considered negligible, due to the minor amount of shortening and thickening of the crust (Porkoláb et al., 2023). Furthermore, models could be improved and validated by incorporating vitrinite

reflectance data from wells, but this option has not yet been implemented in the modelling workflow.

### 5.2 Implications for the thermal evolution of the lithosphere

Royden et al. (1983) suggested that the elevated heat flow and geothermal gradient in the Pannonian basin can only be explained if the mantle lithosphere attenuation was more pronounced than crustal stretching (β>δ). Crustal and subcrustal stretching factors calculated by Lenkey (1999) largely support this finding, while they predict large variations in subcrustal

stretching in the study area, extending from β =1 in the Balaton Highland to β=3.5 in the Zala basin. Predicted subcrustal stretching in this study for the same area represents generally higher β values between 2.5-6 (Fig. 5b). Posterior β values are generally higher in basins (Zala basin, Danube basin) than basin margins. The estimated subcrustal stretching is highest in the Zala basin (up to ~6), while β is slightly lower (~5) in the Danube basin. This does not necessarily mean that lithosphere thinning was less pronounced but can also be due to the fact that extension in the NW part of the study area happened earlier

compared to the Zala basin and Transdanubian Range (Šujan et al., 2021; Fodor et al., 2021). Lower predicted β values in the Danube basin can simply mean that the thermal relaxation of the lithosphere is in a more advanced stage here, due to the older main stretching phase that is not considered in the model.

Using these crustal and subcrustal stretching factors for mantle lithosphere extension between 18-10 Ma, together with accounting for the thermal effect of sedimentation and changes in upper crustal heat generation, we were able to reproduce present-day temperature observations representing a conductive thermal regime. It must be noted that the predicted subcrustal stretching might not be entirely correct due to changes in the timing of stretching throughout the study area as well as further model limitations (section 5.1) but provide a realistic picture for the degree of lithosphere attenuation for the selected input parameter combinations.

The moderate lateral variations in modelled past and present-day lithosphere temperatures (Figs. 8, 9) and β field (Fig. 5) suggest that the lateral variations in the past and present-day lithosphere thickness are rather limited in the study area. This agrees with the LAB depth recently inferred from seismological observations (Kalmár et al., 2023), with predictions between ~60-80 km in the study area (dashed purple lines in Fig. 8 based on Kalmár et al. (2023)). Previous LAB depth maps (Horváth et al., 2006; Tari et al., 1999) infer significantly higher values up to ~105 km in the NW part of the study area, while these were constructed based on limited seismological data derived from lower number of seismic stations compared to Kalmár et al. (2023). Lithosphere scale thermal models of Lenkey et al. (2017) and Békési et al. (2018) building on the previous LAB depth map may therefore predict inaccurate temperatures deep down in the lithosphere in NW Hungary. We compared the present-day posterior model with one of the temperature models of Békési et al. (2018) incorporating the thermal footprint of extension without actual transient calculations. Lithosphere temperatures below ~ 10 km depth in Békési et al. (2018) are significantly higher than in case of the current model, suggesting that steady model assumptions to mimic transient thermal processes led to the overestimation of deep lithosphere temperatures. The predicted post-extension temperature field generally shows a similar trend of evolution as previous studies (e.g. Balázs et al., 2021; Majcin et al., 2015), although direct comparisons with these models were not made due to the different input parameters, modelling approaches, model presentations and timing of modelled temperatures.

In terms of the shallow (<5 km) temperature field, predicted temperatures in the Danube basin and Zala basin are generally in the range of those presented in Lenkey et al. (2017) and Lenkey et al. (2021), while slightly lower than the conductive thermal model predictions in the OGRe database (Ogre, 2020). Additionally, higher lithosphere thickness adopted in Lenkey et al. (2017) in the Western periphery of Hungary, discussed in the previous chapter, might be partly responsible for the lower predicted temperatures also in the shallow sedimentary units of the western periphery of the study area. Our thermal model assumes a conductive thermal regime, and therefore cannot be considered reliable at areas where groundwater flow in fractured/karstified carbonates possibly influence/dominate the temperature field. Although, deeper down in the lithosphere, we consider past and present-day conductive temperature predictions realistic.

## 5.3 Rheological inferences of the new thermal model

Temperature substantially influences the rheology of the lithosphere, as the ductile strength of rocks is an exponential function of temperature. The transient thermal model presented here is significantly more realistic below ca. 10 km depth with respect to previous models (Békési et al., 2018; Limberger et al., 2018), hence, it allows a more precise evaluation of lithosphere rheology. We estimate the yield stress (maximum differential stress prior to frictional or ductile yielding) of the lithosphere by the combination of Byerlee's law for frictional deformation and dislocation creep flow laws for the upper crust, lower crust, and mantle (for material-dependent parameters see Table A1). For the upper crust, we use Westerly granite flow law (Hansen and Carter, 1983). For the lower crust, we use a 0.7-0.3 mixture of mafic granulite and dry quartz (Ranalli, 1995) according to the typical composition of the lower crust (Török, 2012). To calculate the material constants of the mixture, we apply the formula of (Tullis et al., 1991). For mantle creep, we use a wet olivine average from Ranalli (1995) and Kirby and Kronenberg (1987). Strain rate is defined as an average value for NW-Hungary (3 nanostrain/yr), based on Porkoláb et al. (2023). For Byerlee's law, we use a coefficient for compression ( = 3) based on Ranalli and Murphy (1987), a pore fluid factor for hydrostatic case ( = 0.36) and the gravitational acceleration constant (9.81 m/s$^2$).

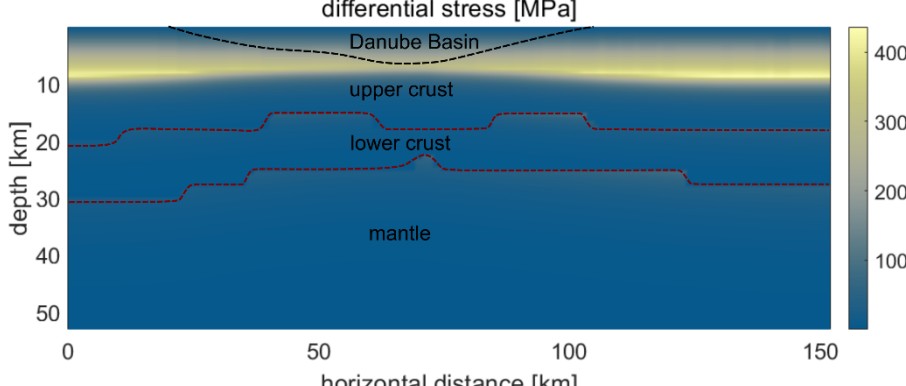

**Figure 10: Differential stress (yield stress) profile from the Sopron Mts. (left) to the Balaton Highland (right, for map-view trace see Fig. 3b). Black dashed line indicates the lower limit of the Danube basin, red dashed lines indicate the Conrad and Moho discontinuities based on Kalmár et al. (2021).**

Results show that most of the lithospheric strength is concentrated in the shallow parts of the upper crust, which is the only brittle layer in the Pannonian lithosphere (Fig. 10). Increased differential stress levels below this shallow upper crust are possible at discontinuities (such as the Conrad and Moho) where lithology and thus viscous creep parameters change. The brittle-ductile transition zone is marked by a sharp decrease in differential stress (Fig. 10) at 7-8 km depth below the Danube basin and 9-10 km at the basin margins (Sopron Mts and Balaton Highland), showing that basins are relatively weaker, and viscous creep in the upper crust becomes efficient at quite shallow levels due to high temperatures. These rheological estimations agree with the generally shallow, < 12 km depth of earthquake hypocenters in the Pannonian basin (Tóth et al., 2002-2010; Lenkey et al., 2002; Porkoláb et al., in press; Bondár et al., 2018). For a more detailed analysis and parameter test see Porkoláb et al. (in press).

## 5.4 Geochemical implications

The lithosphere-scale thermal model is of high relevance to decipher the structure of the lower lithosphere via the

understanding of the vertical distribution of upper mantle-derived rocks. In areas like the Bakony-Balaton Highland Volcanic Field (BBHVF) in the Balaton Highland (Fig. 1.), where hundreds of xenoliths have been described representing the subcontinental off-cratonic lithospheric mantle, is an ideal example. Over the past 20+ years, extended petrographic, geochemical and deformation knowledge has been gained via the detailed investigation on these mantle xenoliths from this region (Szabó et al., 2004). Xenoliths have been brought to the surface by intracontinental monogenetic basaltic volcanism

between ~8 to 2.6 Ma (Balogh et al., 1986; Balogh and Nemeth, 2005; Wijbrans et al., 2007), thus xenoliths from the same eruption event represent lithospheric mantle portion with the same age than the upbringing volcanism. Considering this scenario, sampled subcontinental mantle lithosphere is available for the time slices of 7.96, 4.53, and 2.61 million years (based on Ar-Ar dating on the pyroclastic rocks by Wijbrans et al. (2007) of Tihany, Szigliget, and Füzes-tó volcanoes, respectively), similar to the ages to of the thermal models in this study.

Subcontinental lithospheric mantle, stable between Moho and LAB discontinuities, mainly consists of Mg-Fe-Ca silicates like olivine, ortho- and clinopyroxene. In addition to the silicates, Al-bearing phases such as garnet and spinel yield the rock stability at higher and lower pressure (depth), respectively. In the study area, as a result of the extremely thinned sampled continental lithosphere, only spinel-bearing rocks (lherzolites) have been documented among the mantle-derived xenoliths. For the garnet-bearing mantle xenoliths (sampled at rather cratonic lithospheric portions), mineral chemistry-based pressure

(depth)-temperature relations of the lithospheric mantle can be applied to understand the structure of the mantle lithosphere (O'reilly and Griffin, 2010). In contrast, for the spinel lherzolite-type rocks, only temperature calibration can be used based on orthopyroxene-clinopyroxene mineral equilibrium (Brey and Köhler, 1990) owing to the lack of any geochemistry-based pressure or depth estimation. The equilibrium temperature of BBHVF mantle xenoliths is between 880 and 1160 ± 16 °C (Szabó et al., 2004).

The temperature model provided in this study can overcome this issue by the fruitful interdisciplinary application of petrologic, geochemical and geophysical tools. This approach may give plausible estimation for the depth of origin of the mantle xenoliths. It is important as the question whether or not the mantle xenoliths derive from specific depth(s) or are well distributed for the entire mantle lithosphere remains unanswered for most of the mantle xenolith locations worldwide. Sampling depths of mantle xenoliths from the study area were calculated by crossing of the suitable geotherm (i.e., model age of the thermal model and

volcanic eruption age should be the closest possible) with the isotherm derived from the aforementioned mineral equilibrium. Using these data, the following sampling depth range were provided: Tihany - 41-61 km (23 sample; sampling age and thermal model age: 7.96 and 8.00 Ma, respectively), Szigliget - 39-66 km (25 sample; 4.53 and 4.00 Ma) and Füzes-tó - 36-70 km (72 sample; 2.61 and 2.00 Ma). It is noteworthy to mention that depth ranges show continuous distribution between the shallowest and deepest depths. Approximating from present-day Moho and LAB depth of the study area (Kalmár et al., 2023), we can

thus state that most of the mantle lithosphere has been vertically sampled in the tested three time slices. In other words, using

the new thermal model on mantle xenolith datasets, we could test and confirm their representativity for the mantle lithosphere volumes.

## 6 Conclusions

The presented methodology of incorporating transient thermal effects, using crustal and subcrustal stretching factors, and accounting for sedimentation proved successful in reproducing the most important thermal footprints of basin evolution. The extension of the forward model with the inversion workflow to condition the model with temperature observations provided quantitative measures for the reliability of the models and allowed to constrain model parameters. Further model uncertainties resulting from the selection of model input parameters were investigated through a sensitivity analysis. Additional model limitations and assumptions that add to the overall uncertainties of the modelled (deep lithospheric) temperatures and stretching factors are discussed, to provide a more complete picture of model uncertainties. Past and present-day temperature predictions for NW-Hungary can be considered realistic within the whole lithosphere, while it should be noted that the predicted thermal field and stretching factors are valid for the specific case of input parameters. The calculated crustal and estimated subcrustal stretching values indicate that 1) subcrustal stretching was indeed much more important than crustal stretching in the Pannonian basin: at least half of the mantle lithosphere through the study area was attenuated; 2) subcrustal stretching affected the study area with higher degrees compared to crustal stretching, the crust at several marginal areas remained (almost) intact while crustal thickness under basins decreased to more than half of the assumed pre-stretching setting. These findings generally agree with expectations such as the rise of the asthenosphere translates to larger-scale ductile deformation of the lower part of the lithosphere, while the extension through faulting in the brittle (upper) crust is more localised. Additionally, the predicted present-day lithosphere temperatures suggest that the depth of the current LAB is relatively homogenous, supporting the new seismological model of Kalmár et al., 2023. The new temperature model allows the improved estimation of lithosphere rheology and the origin of mantle xenoliths over the Balaton Highland. The presented methodology can be adopted and applied to model the thermal evolution of sedimentary basins worldwide. The resulting past- and present-day temperature predictions can further be used to constrain geodynamic processes of the study area and provide first-order input for geothermal exploration.

## Appendix A: Model sensitivity analysis

| Model name | Initial lithosphere thickness | Initial crustal thickness | beta | RMS |
|---|---|---|---|---|

| | | | | |
|---|---|---|---|---|
| Model 1 (18 MA) | 120 | 35 | - | Not applicable |
| Model 2 (18 MA) | 135 | 40 | - | Not applicable |
| Model 3 (18 MA) | 150 | 45 | - | Not applicable |
| Model 1a (0 MA) | 120 | 35 | 2 | 2.23 |
| Model 1b (0 MA) | 120 | 35 | 3 | 1.79 |
| Model 1c (0 MA) | 120 | 35 | 4 | 1.5 |
| Model 2a (0 MA) | 135 | 40 | 2 | 2.28 |
| Model 2b (0 MA) | 135 | 40 | 3 | 1.83 |
| Model 2c (0 MA) | 135 | 40 | 4 | 1.53 |
| Model 3a (0 MA) | 150 | 45 | 2 | 2.33 |
| Model 3b (0 MA) | 150 | 45 | 3 | 1.88 |
| Model 3c (0 MA) | 150 | 45 | 4 | 1.61 |

**Table A1. Overview of the sensitivity analysis for initial crustal and lithospheric thickness and subcrustal stretching factors and the resulting model errors (RMS).**

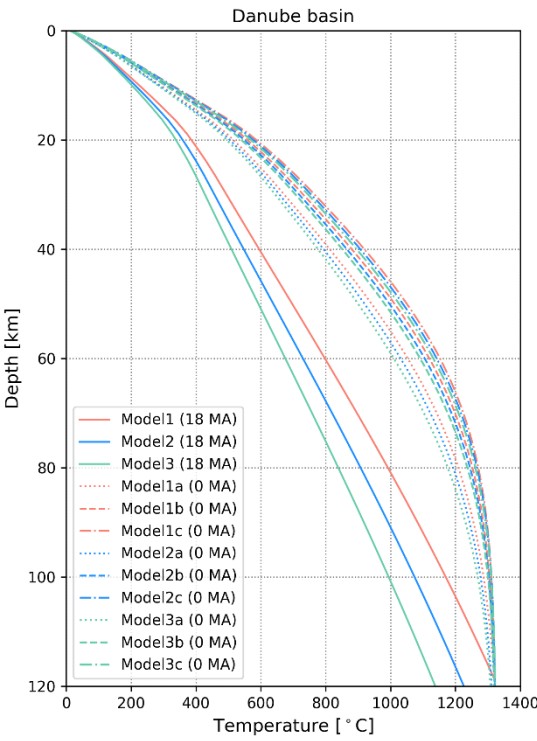


**Figure A1. Resulting temperature profiles of the sensitivity analysis, at the location of the Danube basin (for location see Fig. 3).**

**Appendix B: Material properties for differential stress (yield stress) calculations**

|  | sediments | upper crust | lower crust | mantle |
|---|---|---|---|---|
| $\rho$, density [kg/m³] | 2500 | 2650 | 2850 | 3300 |
| $n$, power law exponent [-] |  | 3.3 | 3.55 | 4 |
| $E$, activation energy [kJ] |  | 186.5 | 340.8 | 471 |
| $A$, pre-exponential constant [Pa$^{-n}$.s$^{-1}$] |  | $3.16*10^{-26}$ | $3.01*10^{-21}$ | $2*10^{-21}$ |


**Table B1. Material properties for differential stress (yield stress) calculations (Fig. 7). Upper crust and sediments: Westerly granite (Hansen and Carter, 1983). Lower crust: 0.7-0.3 mixture of mafic granulite and dry quartz (Ranalli, 1995) Mantle: wet olivine average from Ranalli (1995) and Kirby and Kronenberg (1987).**

## Data Availability

Temperature models have been deposited in Mendeley with the primary accession link https://data.mendeley.com/drafts/vp7jdp79y4.

## Author contributions

**Eszter Békési**: Conceptualization; Investigation; Methodology; Validation; Visualization; Writing - original draft; Writing - review & editing; **Jan-Diederik van Wees**: Conceptualization; Methodology; Resources; Software; Supervision; Validation;

Roles - original draft; Writing - review & editing; **Kristóf Porkoláb:** Conceptualization; Visualization; Roles - original draft; Writing - review & editing; **Mátyás Hencz:** Writing - review & editing, **Márta Berkesi:** Conceptualization; Investigation; Project administration; Resources; Roles - original draft; Writing - review & editing

## Competing interest

The authors declare that they have no conflict of interest.

**Acknowledgements**

The reported investigation was financially supported by the MTA FI FluidsByDepth Lendület (Momentum) project, provided by the Hungarian Academy of Sciences (grant nr. LP2022-2/2022) and by the National Research, Development and Innovation Fund, Hungary under grant number PD147116. Figures were created with Qgis, Inkscape and Python, using the scientific colour maps (when applicable) of Fabio Crameri (http://doi.org/10.5281/zenodo.1243862).

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
