# Peer review of "Modelling the thermal evolution of extensional basins through lithosphere stretching factors: application to the NW part of the Pannonian basin"

_EGUsphere, 2024_

## Author Comment (AC1)

Dear László,

Thank you for reviewing our manuscript, providing constructive comments as an expert both in thermal modelling and in the study area. We incorporated most of the suggestions, which significantly improved the models and the whole manuscript. Most importantly, the revised thermal conductivities we applied provide more realistic input for the models. We also tested the effect of selected input parameters on the model and provided an estimate on the uncertainties, including both a quantitative (resulting from the inversion) and qualitative (discussion on the effect of model assumptions and fixed input parameters) assessment. The temperature predictions have slightly changed in the revised models, together with the estimated amount of lithosphere extension, due to the revised input parameters we applied. Please find our detailed point-by-point responses to the comments below.

Kind regards,

Eszter and co-authors

REVIEWER#2 László Lenkey

*Dear Eszter and coauthors,*

*The manuscript about modeling transient thermal processes in the lithosphere in the NW part of Hungary presents a method to assess deep lithosphere temperatures. The transient conductive heat transport equation is solved, and the calculated temperatures are fitted to observed temperature data to constrain the subcrustal stretching factor. The transient model considers the two most relevant processes, which were active during the evolution of the area: lithospheric stretching and sedimentation. The modeled lithosphere temperatures are used to deduce rheological inferences and estimate the depth of origin of mantle xenoliths found in the region. It is a valuable manuscript, but I suggest modifications and clarifications before publication.*

*The past and present temperatures are calculated by solving the transient conductive heat transport equation (Eq. 2). The calculated temperature depends on the initial conditions, the thermal parameters and the vertical velocity vz. You fix these quantities except the vertical velocities related to stretching, thus the results are valid for this specific model. Other choice of the quantities probably would result different stretching factors and different lithosphere temperatures. In the following I will discuss the effects of thermal conductivity of sediments and the sedimentation rate.*

*The thermal conductivity (TC) of sediments in the model ranges from 1.2 W/mK to 2 W/mK (Table 1). These values are lower than the ones measured on clastic sedimentary rocks from Hungary (ranging from 1.5 W/mK to 5 W/mK in Dövényi and Horváth, 1988; Dövényi et al., 1983, shales: 2.3 ± 0.56 W/mK, sandstones: 3.75 ± 0.71 W/mK summarized in Mihályka et al. 2024). Based on the measured TC values Dövényi and Horváth (1988) established TC-depth trends for shales and sandstones, which result in higher TC than used in the study. The TC of*

*carbonates is also lower than the measured ones. For conductivity of carbonates in Hungary see Dövényi et al. (1983). (limestones: 2.7 W/mK, dolomites: 4.4 W/mK).*

We thank the reviewer for pointing out this mismatch. We revised the calculation of thermal conductivities in the model, and the resulting ranges in sediments are now higher, but still lower than the average values reported by Mihályha et al. (2023), although the matrix thermal conductivity values by Dövényi and Horváth, 1988 were used for the individual lithotypes. This can be explained by the composition of the sediments, where the ratio of sales compared to sandstones in the deep compacted sediments is high. We also revised the thermal conductivities of carbonates based on Dövényi et al. (1983). For more details and discussion, we refer to the revised ms.

Indeed, the selection of TCs, together with the revision of the initial crustal and lithospheric thickness significantly influenced the modelled temperatures, resulting also in a revised posterior subcrustal stretching pattern and magnitude. We emphasise in the manuscript that the resulting solution is valid for this specific case of input parameters and test the effect of variations in several input parameters. For more details, please see the revised ms.

*As the model temperatures are fitted to the observed values, we would expect lower heat flow using the model TC's compared to the observed heat flow. You provided the model results and the thermal parameters in an asset for reviewers, and I calculated the model heat flow in the depth interval 1000-1200 m (Fig R1). The modeled heat flow is uniform in the area: 70 ± 3 mW/m2. Except the Zala basin and part of the Danube basin, where the observed values are 90 mW/m2 and 85 mW/m2, respectively, the modeled heat flow is close the to the observed one (disregarding also the Transdanubian range, where groundwater flow occurs). Better fit to the heat flow could be achieved by varying the TC, heat production and subcrustal stretching factor, but it was not the purpose of the study as you mentioned in the manuscript.*

This is a valuable suggestion to further vary the TC and heat production to achieve a better fit with observed heat flow, which we will consider for further modelling studies that concentrate on the shallow crustal temperature field.

[Figure]

Fig. R1. Comparison of observed and model heat flow.
Colored map: observed heat flow (Lenkey et al., 2021), isolines model heat flow.

*In Eq. 2 constant sedimentation rate is applied. Over the center of the basins, where sedimentation took place in Quaternary, the constant sedimentation rate is a good approximation. In the peripheral parts of the basins erosion has been taken place since Pliocene due to basin inversion and uplift. (See e.g. the seismic sections published in Szafián et al. (1999), a paper you refer.) Erosion increases the subsurface temperature thus, in the peripheral parts less lithospheric stretching is required to obtain fit to the observed temperatures.*

We thank the reviewer for this suggestion, we added a discussion on the effect of assuming a constant sedimentation rate and neglecting the direct effect of erosion, as well as the incorporation of the effect of basin inversion. Please see also the responses to reviewer #1 and the revised manuscript.

*As it is demonstrated varying the TC of sediments and sedimentation/erosion rate would change the modeled T and heat flow. The question arises: how much is the uncertainty of the derived stretching factors and the calculated deep lithosphere temperature?*

*Please make an estimate of the uncertainty of the stretching factor and the lithosphere temperature.*

We performed a sensitivity analysis of initial crustal and lithosphere thickness as well as prior subcrustal stretching values (Appendix A). We also added a section "Model uncertainties and limitations" to the discussion. The standard deviation of beta values reported in https://data.mendeley.com/drafts/vp7jdp79y4 (the dataset will be available with the publication of the paper.) provides a qualitative estimate of uncertainty in relationship of observed temperatures and subcrustal lithosphere model effects. Please note that the LAB uncertainty cannot be assessed by the method and would add to the standard deviation of beta values. For more details, we refer to the revised ms.

*My questions and notes related to the text are the following.*

*Lines 107-108 The sediment bulk thermal conductivities were finally obtained using the geometric mean of the bulk matrix conductivities and the thermal conductivity of the pore fluid.*

*How much was the porosity?*

In the case of shale and sand, we used the porosity-depth trends for the Pannonian Basin by Szalay (1982) to derive compaction coefficients and estimate the porosity. For conglomerate and marl, typical values reported by Hantschel and Kauerauf (2009) were used. We also added this description to the revised ms.

*Line 148 outflow temperatures were marked by uncertainties of ±5 ºC,*

*Do you mean outflow temperature as temperature at the well head? Well head T is unreliable, the error can be much more than ±5 C.*

We increased the uncertainty of outflow temperatures to ±10 ºC.

*Line 184 What kind of numerical method did you use to integrate Eq. 2?*

*The last term in the equation is a partial derivation.*

For the transient numerical modelling of the temperature evolution of equation (2), a 3D explicit 3-step Runge-Kutta finite difference approach was used, with a finite volume approximation and adaptive timestepping. We also added this to the revised ms.

*Line 211 Eq.3 is mistyped. It is correctly: (Zcrust init- Zbasement )/ZMoho present.*

Corrected.

*Line 259 temperatures up to 170 ºC, meaning a geothermal gradient of ~45 ºC/km*

*gradT=(170-12)/4=39.5 C/km*

Corrected.

*Lines 262-263 Negative anomalies can be attributed to outcropping/near-surface basement rocks (mostly carbonates) having significantly higher thermal conductivities*

*Near Sopron and Rechnitz the lower temperature is partly caused by lower lithospheric stretching relative to the basin areas.*

We extended the ms. with the effect of lower stretching compared to basins.

*Lines 264-265 The conductive assumption is although not fully valid for parts of the Transdanubian range built up by fractured and karstified carbonate rocks.*

*A larger area than the Transdanubian rage is affected by groundwater flow and convective heat transport, because groundwater flow also occurs in the carbonate rocks covered by sediments.*

We thank the reviewer for this suggestion that we incorporated in the model. We further limited the use of temperature observations in the vicinity of the Transdanubian Range based on the shallow temperature maps of Lenkey et al., 2021 to include measurements from areas that are potentially influenced by fluid flow, resulting in the reduction of temperature observations to 319. We also added to the text that fluid flow in buried carbonates can also occur, influencing the thermal field. For more details, we refer to the revised ms.

*Line 270, Fig. 6. The grey color code to visualize the difference between model and observed temperatures is not suitable to quantify the difference.*

Corrected.

*Lines 322-324 Towards the Transdanubian Range (Balaton Highland), predicted model temperatures are slightly higher in the deeper part of the model compared to the NW part*

*(Sopron Mts.). This might be explained by the shift in the timing of active rifting, that migrated from NW towards SE (e.g. Balázs et al., 2016).*

*The rifting time was fixed in the model, so the temperature difference has a different reason, e.g. different subcrustal stretching factors.*

We revised the text and excluded this paragraph.

*Line 329 we compared the overall misfit between modelled and observed temperatures*

*How did you calculate the difference between the modeled and observed temperatures as they belong to different depths?*

Only one observation per grid cell was supported in the models, so observations were restricted to 200 m deep intervals, and measurements with lower uncertainties were considered. We restricted the calculation of misfits also to the model resolution. We added the description of one observation per grid cell to the text in the methodology section. Please see the revised ms.

*Lines 349-351 It must be noted that the predicted subcrustal stretching might not be entirely correct due to changes in the timing of stretching throughout the study area but provide a realistic picture for the degree of lithosphere attenuation.*

*It is not only the timing of stretching, which influences the stretching factors, but all parameters used in Eq. 2.*

This is an important point; we also discussed the influence of other parameters in the revised ms.

*Lines 366-368 These differences in shallow temperature predictions can partly be explained by the different calibration datasets used by Lenkey et al. (2017) and (Lenkey et al., 2021), excluding temperature measurements from (recent) geothermal wells documented in the OGRE database.*

*OGRe (2020) is a very useful database to get quick-look temperature data. However, no information is given about the conditions of the measurement. E.g. it is not known if the BHT value is corrected or not. In the Geothermal Database of Hungary (Dövényi, 1994, Lenkey et al., 2021) the observed data are corrected if possible, and every temperature data is quality checked, and depending on the type and conditions of the measurement they are ranked into quality categories.*

We revised this sentence in the ms. and we are aware that the quality of the two datasets is not comparable. We also revised the uncertainty of outflow temperatures mostly reported in OGRE to account for larger uncertainties, as described in the previous comments.

*Lines 398-399 give references to Porkoláb et al.*

Reference updated

---

## Author Comment (AC2)

Dear Reviewer,

We would like to express our gratitude for the thorough review of our work. We received many insightful comments that have helped us to significantly improve the manuscript. We revised the model input parameters and performed a sensitivity analysis to a selection of model input parameters. The temperature predictions have slightly changed in the revised models, together with the estimated amount of lithosphere extension, due to the revised input parameters we applied. Please find our detailed point-by-point responses to the comments below.

Kind regards,

Eszter Békési and Co-authors

REVIEWER #1

*The manuscript by Bekesi et al. entitled 'Modelling transient thermal processes in the lithosphere: application to the NW Pannonian basin' presents a simplified modeling study on the thermal evolution of the NW part of the extensional Pannonian basin considering distinct crustal and mantle thinning factors and sedimentation. The calculated new thermal field is then used to present a 2D yield stress section of the lithosphere. Finally, the manuscript contains a brief discussion on mantle xenoliths. Given the large number of major issues of the manuscript, I suggest substantial revision before considering it for publication.*

*The title does not reflect the content of the manuscript. Reconstructing the thermal evolution of the lithosphere and particularly the deep lithospheric mantle is challenging, indeed, because of the large number of transient effects, i.e. partial melting, melt emplacement, phase changes, shear heating, non-uniform upper crustal, lower crustal and mantle thinning, paleo-surface temperature variations, basin inversion and related deformation, water circulation, etc. This manuscript uses the stretching factors approach of Royden and Keen (1980) to somehow consider crustal and mantle thinning in a simplified way, but none of the other transient effects are taken into account.*

We thank the reviewer for highlighting also the transient processes not considered in the modelling technique. We revised the title to reflect the methodology more clearly. Still, the methodology is capable of reconstructing the most important transient effect that accompany extensional basin formation through the crustal and subcrustal stretching factors and sedimentation, as well as the detailed present-day crustal geometry. We extended the manuscript by commenting on the transient processes not taken into account in the modelling and discussing their potential effects. For more details, we refer to the revised ms.

*The abstract and the manuscript claims that one of the main goals is to better quantify the thermal field in the entire lithosphere. There are too problems with this: (1) the model does not use any observational constraints from the deep basins, crust or lithosphere, and likely it is not sensitive to temperature variations at great depths; therefore, the goal cannot be reached with this method. (2) While the manuscript presents one possible model result, a sensitivity analysis,*

*assessing the role of different initial and boundary conditions and input parameters are missing, therefore, it is not an easy task to see how robust or reliable is the model. No model limitation section is included, despite the large number of assumptions the authors made.*

We agree that the limitations of the model have to be highlighted and the fact that the model is valid for a specific case of input parameters should be discussed. We revised the ms. with the detailed description and validation of input parameters we used. Additionally, we performed a sensitivity analysis to test the effect of variations in input parameters. We also included a model limitation description. About deep crustal and lithospheric constraints; the model considers a detailed present day crustal geometry, including the crustal thickness map based on seismological observations (Kalmár et al., 2021) and is compared with the present-day lithosphere thickness (Kalmár et al., 2023). For more details, we refer to the revised ms.

*The used model parameters: Many parameters are not justified and seem to be far from reality. The source of other input data is not clarified and thus cannot be checked. A model needs to be reproduceable by the community and you need to make available the most important input data. 1. Initial crustal thickness: the authors assume a constant 35 km thickness. What is the source of this parameter? The study area includes metamorphic core complexes, their formation requires a thick and hot crust, which infers that your chosen initial values are lower than it should be. Previous reconstructions (e.g. van Hinsbergen et al. 2020) reported a much larger amounts of extension and a thicker initial crust. Geochemical studies based on xenoliths inferred a much larger initial crustal thickness (Torok et al.). Finally, the basement units of the region derived from the Alps, likely having a much thicker crust than 35 km in the Early Miocene. The initial lithospheric thickness of 120km: what is the constraint on this and how much role does it have? Lithologies: what is the source of this? For instance, the sand to shale ratio is proposed to be 1:9 for the 'Lower Pannonian' (Upper Miocene). How is this constrained? After a brief google search, well logs published by Stano et al. 2016 shows a sand to shale ratio of at least 50%. This means that your applied thermal conductivities are wrong, and this is a major issue. The timing of extension: in the model a uniform timing for rifting is assumed between 18-10. Most structures are inferred to be active only until the Middle Miocene (e.g. Majcin et al. 2015), a few small-offset normal faults would not have influenced lithospheric thinning.*

We revised the ms. to better describe the input data and parameters we used. First, we revised the initial crustal and lithospheric thickness to 40 km and 135 km, respectively, to better represent the overthickened pre-extension lithosphere of the region. We also tested the effect of a range of input values as described in the previous comment.

We corrected the sand to shale ratio of lower pannonian sediments to 30:70. The ratio of sand is indeed even higher in some wells shown in Sztanó et al., 2016, but the thickness of Lower Pannonian sediments dominated by clay is relatively large in the Danube basin and Zala basin compared to the sandier turbidites (Szolnok fm.). We finally chose the Bősárkány-1 well to define a realistic ratio of 30:70, which we considered an acceptable average for the whole study area. For more details and references, please see the revised ms and further replies to the specific comments on thermal conductivities of reviewer #2.

The crustal thinning factor is simply calculated from the present-day crustal thickness, basement depth and initial crustal thickness (eq. 3.). For the initial crustal thickness, we assumed 35 km in the previous model, which was updated to a more realistic value of 40 km, but an initial thickness of 45 km was also included in the parameter test. The higher initial crustal thickness results in higher thinning factors (~ 1.3) also in the Rechnitz core complex area. The initial crustal thickness of 40 km was chosen as an input value that is realistic for the majority of the study area, however, it is important to note that the crustal thickness was possibly even larger in the western periphery of the study area. Higher initial crustal thickness in e.g. the Rechnitz core complex area would have resulted in even larger crustal thinning factors. We included this discussion in the revised ms.

We used a uniform timing for rifting for simplicity (to avoid inverting for subcrustal stretching factors for different times by introducing further parameters in the inversion complicating the models), that covers the main rifting phase for all parts of the study area. We agree that rifting was no longer intense after the Middle Miocene in parts of the study area, although active rifting e.g. in the Transdanubian Range is younger (~15-8 Ma, Fodor et al., 2021). We tried to choose a rifting period that covers the period of the intense rifting phase of all parts of the study area as described in the revised ms. We also comment on this limitation and potential effects of uniform timing of rifting on the resulting stretching factors in the ms.

*The model result: this is already a mixture of discussion and describing some results. How is it possible that nearly 0 crustal thinning is calculated for the Rechnitz core complex area, that must have undergone substantial crustal thinning? This is a sign of the wrong model parameters. In Figure 6, it is not possible to read the values of the mismatch between the well and model data, but it still seems to be a significant error. About sediment blanketing: in the results of the shallow temperature field chapter you write: "Positive anomalies are the reflection of sediment blanketing, meaning the insulating effect of sediments with low thermal conductivity." – The deposition of cold sediment would lead to decreased temperature values at shallow depth and higher thermal values in the basement because of the blanketing of low conductivity sediments.*

We thank the reviewer for the suggestion to separate the results and discussion sections, but we think the explanation of some basic features and processes can remain in the results part. Otherwise we would need to describe the results again in the discussion part to be able to discuss the observed temperature anomalies. Still, we moved some parts of the results to the discussion. We revised the sentence on the effect of sediment blanketing to be more clear. It is right that cold sediments would lead to decreased temperatures, but post-rift sedimentation initiated 10 Myrs ago, allowing sufficient time for the deposited Pannonian sediments to warm up and have an insulation effect on the deeper/older sediments and basement rocks. About the Rechnitz core complex, please see the previous reply. We also revised the representation of mismatch between modelled and observed temperatures. In terms of the magnitude of errors between modelled and observed temperatures, we show the largest mismatch on the maps. These values may remain relatively large due to potential measurement errors but may also be attributed to local variations in sediment geometry and composition that are not captured by the model, or can even be caused by local fluid convection e.g. in the carbonate basement as detailed in the ms.

A better fit with the overall temperature measurements at shallow depth could have been achieved with introducing variations in the thermal conductivity of sediments, accounting for local TC anomalies because of compositional differences (i.e. sand:shale ratio) compared to the average values we used, but the precise representation of the shallow (<5 km) thermal field was not the main purpose of this study. We discuss this also in the further points of reviewer #2.

*How did you consider the uplift of the basin margins linked to the ongoing inversion of the basin (e.g. Bada et al. 2007)? Likely it would have a major impact.*

The geometry and structure of the uplifted basin margins are taken into account by the present-day crustal geometry that is used as a model input. Further effects of the neotectonic inversion on the temperature field are considered negligible, due to the minor amount of shortening and thickening of the crust. Based on present-day shortening rates in the NW-Pannonian basin, the accumulated shortening strain over 8 Myrs would be around 0.024 (strain rates based on Porkoláb et al., 2023), which is around 5 km over a 200 km long section. This estimation shows that the thermal effects of this shortening is probably very low, and the most important effect is the geometry of the uplifted basin margins, which we do take into account in the model. For more details, please see the revised ms.

*Structure of the manuscript: The results and their discussion are not separated. You should make clear which parameters and which model outputs are well constrained and what is the sensitivity of others.*

We separated the results and discussion more clearly where applicable and described the model inputs and uncertainties in more detail. For more details, please see the previous comments and the revised version of the ms.

*Comparison with previous studies: this manuscript doesn't even mention previous modelling efforts on the crustal and mantle thinning, surface heat flow and basin temperature evolution. In the detailed comments below, you find many useful papers that can be used to compare your results with previous inferences. Besides well data, vitrinite information is also widely available in the region that should be used to validate such models.*

We thank the reviewer for the suggestions for previous studies. We revised the ms. to compare model parameters and results with further studies where relevant. About vitrinite reflectance data, we could not implement such observations in the model, but we added a short comment on their potential applicability to the ms.

*Because of the many limitations listed above, the final sentences on the new stress envelope or comparison with xenoliths remain elusive and in general they don't really connect with the manuscript. Instead, you should discuss the sensitivity and reliability of the thermal model and compare it with previous inferences and with other similar regions.*

We extended the discussion with a detailed "Model uncertainties and limitations" section and we also performed a sensitivity analysis. For more details, please see the revised version of the ms.

*further detailed comments:*

*Title: it does not reflect the content of the manuscript*

We revised the title.

*Abstract: reliable thermal evolution is not modelled for the entire lithosphere due to the limitations of the modelling approach and lack of constraints*

We added a sentence on the effect of selected initial parameters and the corresponding sensitivity analysis and model limitations to this specific case of parameter selection.

*ln 19-20: not all the sedimentary basins are extensional*

Corrected for the specific case of extensional basins.

*ln 22: Royden and Keen 1980*

Reference added.

*ln 32: most thermo-mechanical models are constrained by observations, e.g. Lescoutre et al. 2019; Heckenbach et al. 2021; many others*

We revised and partly excluded this part of the introduction.

*ln 37: in the upper crust*

Corrected.

*ln 44: sometimes you include Late Miocene, in other places you write Early to Middle Miocene. Which is true?*

Since extension migrated through time in the entire Pannonian basin, here we revised the text simply to Miocene.

*ln 47: how is this inversion stage considered in the model?*

Please see our previous response.

*ln 53: i.e. compositional changes through sedimentation: what does this mean?*

Here we only refer to the changes in thermal properties due to sedimentation, that corresponds to the change of composition of upper crustal structure. We revised the text to be more clear.

*ln 54-56: you should reflect on the large number of previous thermal modelling efforts in the region, including, but not limited to: Lankreijer et al. 1999; Majcin et al. 2015; Bartha et al. 2018; Balasz et al. 2021; Rybar and Kotulova 2023*

We extended the ms. with more comparisons where relevant and possible.

*ln 59: "high precision" - can you elaborate?*

We excluded this term and discussed model limitations and reliability in the ms.

*ln 59: for (not to)*

Corrected.

*ln 73: lower plate with respect to what? Out of context.*

With respect to the Alpine subduction system, as described in the text. We think that this is not out of context.

*fig. 2: what is the sedimentary basin on the right side? Also indicate the orientation of the section.*

We added a description to the ms. to explain the SE limit of the section and added the section orientation to the figure.

*ln 87: justification?*

We revised this value and discussed its effect, please see previous comments and the revised ms.

*ln 91: delete -*

Deleted.

*ln 95: where is this thickness map presented, shown?*

We uploaded the input carbonate thickness map to the corresponding data repository (https://data.mendeley.com/drafts/vp7jdp79y4), which we will make available with the publication of the paper.

*ln 100-101: justification?*

We revised and extended this part of the ms., please see the previous detailed comments and revised ms.

*ln 118-120: rephrase*

We rephased the text.

*table 1: what about paleogene rocks?*

The extent and thickness of Paleogene rocks at the study area is limited and were therefore not separated from the pre-Pannonian Neogene sediments in the models. Please see the revised ms.

*table 1: what is the source of information behind this data?*

We added a detailed description on how we constrained the composition and properties of the layers, including several references. Please see the revised ms.

*ln 128: is it available or the most important data now made available with this manuscript?*

The references for temperature measurements from which we selected the calibration dataset are described in the text. The original datasets are available in the appendix of Dövényi and Horváth, 1988 and in Dövényi et al., 2002 and on the website of the Geothermal Information System (Ogre, 2020). We uploaded the dataset of selected measurements to https://data.mendeley.com/drafts/vp7jdp79y4, which we will make available with the publication of the paper.

*ln 135: meters?*

Corrected.

*ln 136: in fact I cannot see too many wells in the deep basins. Elaborate*

We specified this sentence to the vicinity of the Zala basin.

*Figure 3: scale of the basement depth map?*

We added the scale to the map.

*ln 161: what about different amounts of upper and lower crustal thinning, likely affecting the Rechnitz region?*

The methodology cannot separate the upper and lower crustal stretching factors, but the resulting thermal effect due to the difference between lower and upper crustal stretching is considered by the present-day crustal geometry and composition. Please see also the previous detailed comments on the basin inversion.

*ln173: sensitivity of this assumption? What if the initial lithopsheric thickness was lower or higher?*

We included a sensitivity analysis in appendix A to demonstrate the effect of initial crustal and lithospheric thickness as well as the subcrustal stretching factor. For more details, please see the ms.

*ln 174-175: In this model, when the lithosphere was thinned to ca. 60 km, you had a 60 km depth domain of constant temperature beneath? How reliable is this? Why dont you use a constant heat flow lower boundary condition?*

We thank the reviewer for pointing out the need for the explanation of the model reliability below the present-day LAB depth. For all models, only lithospheric thermal properties were considered, and the post-stretching models extend until the depth of the initial LAB because of the pre-defined model geometry. Model temperatures can only be considered reliable approximately only until the present-day LAB depth. We also added this to the ms. and we describe in the methodology that the post-stretching models have the same geometry and resolution as the starting model.

*Ln 180: so your model is only accurate until 5-10 km depth?*

This statement is not referred to the overall resulting model, only to the steady-state part, and it does not mean the models are not accurate deeper.

*Ln 190: instead of this, it would be more useful to write about the thinning factors of the study area*

We added more discussion on the stretching factors to the ms., but we did not delete this introductory part to help the readers understand the terms used in the modelling.

*Ln 196: grammar*

Revised.

*Ln 198-200: what is the limitation of this?*

We commented on this in the newly added part of the discussion.

*Table 2: Lab: 120 meters?*

Revised.

*Ln 204: There are many other studies calculating different crustal and mantle thinning, e.g.: Lankreijer et al. 1995; 1999; Majcin et al. 2015; Bartha et al. 2018; Balasz et al. 2021; Rybar and Kotulova 2023*

Revised, please see previous comments.

*Ln 205: Primary?*

We deleted this term.

*Ln 206: what does past-extension mean?*

We meant the thermal state after extension, most importantly at present-day. We revised the text.

*Ln 209: why 35 km?*

Revised and explained in previous comments.

*Fig. 5: how would you discuss these patterns?*

We added more explanation to the discussion.

*Ln 259-263: this is discussion, not results*

We agree that this is discussion but it belongs to the interpretation of basic features described and therefore we left the revised version of this sentence in the results part.

*Ln 275: I would respectfully challenge this statement. How can you be sure that the deposition of cold sediments would increase the temperature in such shallow depth? It would*

*increase at larger depth. Of course, you have higher temperature values, where the crust is thinner and therefore the mantle is more elevated.*

Please see our previous comment on the thermal effect of sedimentation. The fact that elevated temperatures in shallow depth are also a result of higher crustal and lithospheric stretching is fully valid and added to the ms.

*Figure 7: which wells are these, what is the source of information? Is it open-source? At least the used and presented well data should be better documented and shared with this manuscript. It is also a warning sign how the errors increase with depth which questions the reliability of the models.*

We added references to the datasets from which we constructed the calibration dataset. We uploaded the dataset used for calibration to https://data.mendeley.com/drafts/vp7jdp79y4, which we will make available with the publication of the paper.

*Ln 294: ref*

Corrected.

*Fig. 10: on the well data the basin was much shallower, which is right? Furthermore, it is not likely that the crust would be laterally homogenous, therefore it is difficult to understand the value of this cross-section. The rheological section would of course be different if heterogeneities were included, but we have no detailed information on these.*

The well is not located along the trace of the cross-section (please see Fig. 3b), therefore the difference between basin depth. We selected a cross-section trace to also include the deepest part of the Danube basin. The section is included to represent the difference between basins and basin margins and to provide an average approximate estimate on yield stresses.

---

## Author Comment (AC3)

Dear Giacomo Medici,

Thank you for providing useful suggestions for the manuscript. We incorporated most of the suggestions, which improved the presentation of modelling procedure, description of results and discussion. Please find our detailed point-by-point responses to the comments below.

Kind regards,

Eszter Békési and co-authors

COMMENT #1 by Giacomo Medici

*General comments*

*It's always good review original paper on large-scale hydro, and thermal models from Hungary! The research is also original and can be exported to many other areas of geothermal interests worldwide. Please, follow my comments to improve the manuscript.*

Thank you for the constructive comments, we improved the manuscript based on the suggestions detailed below.

*Specific comments*

*Abstract*

*Line 10. "The forward model is extended". Please, be more specific. The object of the sentence is unclear and the abstract is short with obvious chance for clarifications*

We extended this sentence to be more specific about the methodology, please see the revised ms.

*Introduction*

*Lines 17-61. Did you consider adding a general statement to steady state and transient modelling in other fields of geo-science? Many large scale (deep and large in plant view) flow models have been developed in the Pannonia Basin. Your country has an original and well recognized academic tradition on this aspect of geo-science.*

We added further references to the introduction. For more details, please see the revised ms.

*Lines 16-20. "Understanding...thermal evolution pattern". Long statement without references. Please, insert recent review papers in the field of geothermal energy for characterization, production and modelling:*

*- Review of Discrete Fracture Network Characterization for Geothermal Energy Extraction. Frontiers in Earth Science, 11, 1328397*

*- Direct utilization of geothermal energy 2020 worldwide review. Geothermics, 90, 101915.*

We thank for the suggestion to include these references. We added other reference examples for the connection between lithospheric thermal field and geothermal energy potential, exploration and exploitation.

*Line 44. Clearly state the other hot basins in Europe (e.g., Rhine Graben, Tyrrhenian Sea). They are not so many and you can avoid vague sentences in that way.*

We extended the text with specifying other hot basins in Europe, such as the Tyrrhenian and Aegean basins with similar settings.

*Line 61. Specify the 3 to 4 specific objectives of your research by using numbers (e.g., i, ii, and iii).*

We extended this part of the manuscript, but we preferred not to use numbering but only describe the objectives in more detail.

*Data and methods*

*Line 127. "We calibrated the thermal model with subsurface temperature measurements from hydrocarbon and geothermal wells". Please, specify the depth of the temperature data used for the calibration. 0.2 - 5.0 km based on geothermal and hydrocarbon observations?*

The depth interval is written later in the text: "200-5100 meters". We will also make available the whole dataset used for calibration with the publication of the paper, together with the modeling results (https://data.mendeley.com/drafts/vp7jdp79y4). For now, availability is restricted to the reviewers.

*Line 127. If we assume observations 0.2 - 5.0 km, did you discuss reliability/validity of the model much deeper? The model should not be very sensitive in the deeper part.*

*Line 127 – onwards. Do you need to add some detail on the sensitivity of your model with respect to the parameters?*

We included a sensitivity analysis and discussed the potential effects of selected model parameters to the resulting temperature estimates. For more details, please see the revised ms and our detailed responses to the reviewers.

*Line 127. Link the depth range of temperature observations to Figure 3a*

It has already been linked to Fig. 3a.

*Line 181-222. The time steps of your transient model should be much more clear when you describe the methodology. They should be clear from the first lines. Do you need a link with the Table 2?*

For the transient numerical modelling of the temperature evolution of equation (2), a 3D explicit 3-step Runge-Kutta finite difference approach was used, with a finite volume approximation and adaptive timestepping. The sentence and reference have been added to the revised ms. We do not think a link with Table 2 is appropriate here.

*Discussion*

*Line 342. "It has already been". Avoid to start a new sentence with "it". Please, revise the language.*

Revised.

*Line 347. "These factors". Difficult to follow. Please, remind the specific factors to the reader.*

We extended the text.

*Line 408. I suggest "considering this scenario". Avoid to use the word "this" alone.*

Corrected.

*Conclusions*

*Line 451. Insert a connector such as "indeed" to link the last two sentences.*

We think the link is already clear here.

*References*

*Lines 477-639. Please, integrate relevant literature as suggested above.*

Other references added.

*Figures and tables*

*Figure 3a. Please, increase the graphic resolution. Some details are difficult to read.*

Corrected.

*Figures 5 and 6. Make the figures larger.*

Corrected.

*Figure 8. Make the letters of the labels larger.*

This would not fit properly to the figure, and we think the labels are readable in full size.

---

## Author Comment (AC4)

Dear Nicolas Coltice,

Thank you for providing useful suggestions for the manuscript. We agree that using the correct terms is important, so we revised the text to use the term inversion instead of data assimilation. We agree that the reported uncertainties resulting from the inversion procedure are not sufficient to represent the overall uncertainties of the model. In the revised ms., we show multiple models to assess the effect changes in input parameters to modelling results and discuss further the influence of assumptions and simplifications we made on the temperature estimates. Please find our detailed point-by-point responses to the comments below.

Kind regards,

Eszter Békési and co-authors

COMMENT #2 by Nicolas Coltice

*The manuscript present a thermal reconstruction study of the Hungarian area of the Pannonian basin. It describes a new inverse methodology in order to obtain tectonic information on lithosphere thinning in the area. First of all, I state here that I am more a specialist of modelling than on the tectonics and geothermics of this area.*

*My point of view is the qualities of the manuscript lie in:*

*- the new methodology employed to get information of the deep lithospheric structure from temperature measurements.*

*- pushing the result towards interpretations on rheology and xenolith depth origin.*

*My opinion is that the shortcomings of this manuscript are:*

*- it is difficult to estimate if the method is able to improve the knowledge of the deep lithospheric structure, especially in a hot and thin crust area in which hydrothermalism and deformation/melting are present. Before inversion, the prior has already a very small misfit (1.33°C). I guess that the uncertainties on the depth of the different layers can introduce such misfit on its own (the thermal gradient is around 40°C/km). The misfit is improved through the process (0.43°C), but is it significant? Since we don't have here an analysis of how varying the properties of rocks and depth of interface within uncertainties impact the mist, it is hard to know if the authors can resolve the deep lithospheric thermal structure. Given the low value of the misfit prior to inversion, I would say no.*

We revised the manuscript to comment further on the uncertainties of input parameters as well as a discussion on the limitations of the model. It is indeed an important point that the misfits prior to inversion were also low in the presented model (we show more scenarios in the sensitivity analysis of Appendix A). Although these reported uncertainties do not reflect the overall uncertainty of the model in the deep lithosphere, from which no direct constraints are available. Therefore, we emphasise in the revised ms. that the temperature estimates are valid

for a specific case of input parameters. Still, we think that the carefully selected model parameters allow for a realistic estimate of past-and present-day thermal field as well as the amount of lithosphere stretching through basin formation.

*- the method is not a data assimilation method. Data assimilation, which is mostly used for chaotic models with butterfly effect, means that there are new data than can be assimilated (correction of the model) in time. Here, the observations are present-day only. So it is a classical inversion problem with a new methodology. This is a detail but it is worth to use the proper terms.*

The used methodology is ES-MDA, which is a data assimilation method according to the Emerick, A. A. and Reynolds, A. C.: Investigation of the sampling performance of ensemble-based methods with a simple reservoir model, Computational Geosciences, 17, 325-350, https://doi.org/10.1007/s10596-012-9333-z, 2013b. We agree with the reviewer that it is used here for inversion for present-day data in agreement with synthetic studies in Emerick&Reynolds, and not progressively updated for incoming data. To clarify data assimilation has been changed to inversion workflow and use of data assimilation has been limited to section 3.4.

*- most of the figures/captions require additional information.*

*Minor details along the text:*

*- Fig.1: explain more clearly why the country borders are used for the study (it is stated later in the text but it would be good to have it here)*

We added the explanation of the study area extent to the caption of Fig. 1.

*- Fig.2: orientation is missing (NW - SE)*

Orientation added to Fig. 2.

*- line 100: how is the LAB defined here? The study is thermal, so it would be good to explain.*

We extended the text with the 1330 ºC prescription of LAB temperature.

*- Table 1: the table is not very informative. Is it possible to either a graph or more details on how the variations are produced?*

We could not find a better way to represent the values, but we extended the description of thermal properties in the text.

*- Figure 3: Why gray, green and black circles for the same information? What does the color mean?*

We revised and simplified the figure.

*- line 146: that would be nice to have more details for the errors. Citing the papers of the first authors does not seem enough to evaluate where they come from.*

We revised the errors associated with the measurements, please also see the previous comment of reviewer #2.

*- line 159: remove statement on the inverse modeling. This is the forward model section and it is fundamental to distinguish the difference between the forward and inverse model.*

We removed the statement and only mentioned inversion referring to the separate section.

*- line 173: why 120km for the LAB?*

Revised and explained in the revised ms.

*- line 178: typo 'preduction'*

Corrected.

*- line 204: what is unrealistic? More details are needed here to evaluate how the authors rule out a model.*

We added a more direct description. The subcrustal stretching factor of only slightly more than 1, predicted for the Transdanubian Range in Lenkey 1999, would results in a thermal field that is almost identical with the pre-extension thermal field, which is unrealistic considering the present-day LAB depth (Kalmár et al., 2023).

*- equation 3: misplaced parenthesis*

Corrected.

*- section 3.4: more theoritical details on the inversion method would be good. Why this one and not another one? Where does equation (4) come from and why is it adapted to the problem?*

We think the provided description is sufficient for an overview, and we refer for more details to the original papers describing the applied inversion procedure.

*- line 248: explain what a variogram is? Provide a figure?*

We added a short explanation to the revised ms.

*- Figure 6: large errors in the hottest spots. Explain please.*

For each well location, the maximum error is reported on the map. We added more description on the potential sources of the misfits in the ms., that can originate from measurement errors as well as local variations in thermal properties and compositions that are not included in the input model.

*- Figure 7: the choice of colors make it difficult to read (black and blue lines especially)*

Corrected.

*- line 294: ref missing*

Corrected.

*- Figure 8: same for colors*

Corrected.

*- Figure 10: what are the units?*

The title of the figure shows that the unit is MPa.

---

## Author Response (AR2)

Dear Editor and Reviewer,

Thank you for the further suggestions to improve on the manuscript. We considered all comments and excluded the chapters on the rheological and geochemical implications. We still write about these (potential) implications shortly in the discussion, but we refer to detailed studies (when available) and clearly state the limitations of the model to constrain deep lithosphere processes. For the additional minor comments, please read our responses below. The comments of the referees are in *Italic*, our reply is in normal font. We have also added a marked-up manuscript version where all the changes in the text can be found.

Kind regards,

Eszter Békési and co-authors

REVIEWER #1

*The revised manuscript of Bekesi et al. addressed almost all my previous comments and this will be an appropriate contribution in Solid Earth. My only remaining major comment addresses chapters 5.3-5.4 about the rheological and geochemical implications. After addressing this one point and considering a few minor comments, the manuscript will be ready for publication.*

*Major comment: Chapter 5.3-5.4: Rheological inferences and geochemical implications: In my opinion, these chapters are still not connected to the rest of the manuscript. The parameters and statements here are not discussed, the limitations behind the assumptions are not provided. As also pointed out by Nicolas Coltice, the thermal model is not particularly sensitive to temperature variations in the mantle and therefore, this model is not suitable for addressing the actual depth of xenoliths. The authors need to choose between two possibilities: they either provide a real discussion behind these topics, including the role of inherited structures in the crust, assumptions on grain size variations, and water content, etc. How realistic is it that the entire mantle is proposed to be described by a wet olivine rheology; what is the possible temperature variation and error within mantle depth in different models, etc. Or these sections should be kept out from this manuscript and could become an interesting independent paper.*

*Further comments:*

1. *Ln. 92: show the location of the "Mid-Hungarian Shear Zone" on one of the maps or in the cross-section.*

   We added the location of the Mid-Hungarian Shear Zone to Fig. 2.

2. *Ln. 125-126: The paper by Faccenna et al. (2014) did not propose that the overriding plates prior to back-arc extension had an overthickened lithosphere. On the contrary, we assume*

*a thick crust and hot lithosphere, such as having a shallow LAB, the isotherm being much warmer than steady-state solutions.*

We corrected the references on the overthickened lithosphere, and limited the assumption based on Faccenna et al. (2014) to the selection of the crustal thickness.

3. *Figures: it is not an ideal choice to plot the "temp. obs." by grey dots on an already grey map*

   Corrected.

4. *Figures: when showing continuous physical fields, including the stretching factors, with the chosen smooth color scales, please add iso-contour lines, otherwise, it is nearly impossible to read the values. This is already very well done in Fig. 9*

   We added isolines to Fig.5, and indicated the temperature values both in Fig 5. and 6.

5. *Ln. 284: pre grid cell?*

   Corrected.

6. *Fig. 6: Given the Range in the southeastern part of the model area is clearly affected by shallow-water circulation and related thermal effects in the porous limestones, I suggest you indicate the boundary of this unit. Furthermore, I suggest also indicting the area of the Rechnitz complex in Figs. 5-6, where the used approximation clearly does not work.*

We outlined the area of the Transdanubian Range built up by outcropping carbonate rocks, affected by shallow fluid flow in Fig. 6. For the Rechnitz core complex, the shallow part of the present-day model (<10 km) can be considered reliable, so we do not show its outline. In the text, we discuss about the validity of model parameters in the Rechnitz area, and we added its outline to Fig. 5, indicating that the prior and posterior stretching factors are not fully realistic in there.

7. *Chapter 5.1: Please also clearly write in this chapter about the fluid flow effects, particularly affecting the Range and the porous limestone reservoirs. Furthermore, you need to also write here that areas, such as the Rechnitz complex were likely affected by different upper and lower crustal stretching factors and the applied method is not suitable to account for such effects connected to metamorphic core complex formation and exhumation.*

We clearly list the most important model simplifications in section 5.1 in the revised ms. We added further discussion on the effects of fluid flow that is relevant to the shallow part of the model also in this chapter. We extended the discussion on the model simplifications and limitations in the peripheral parts of the basin, most importantly in the Rechnitz core complex.

8. *Chapter 5.3-5.4: Rheological inferences and geochemical implications. In my opinion, these chapters are still not connected to the rest of the manuscript. The parameters and statements here are not discussed, the limitations behind the assumptions are not provided.*

*The authors need to choose between two possibilities: they either provide a real discussion behind these topics, including the role of inherited structures in the crust, assumptions on grain size and water content, etc. How realistic is it that the entire mantle is proposed to be described by a wet olivine rheology, etc. Or these sections should be kept out from this manuscript.*

We excluded these sections from the revised ms., and only extended the previous section with a short discussion on the rheology based on Porkoláb et al. (2025). Additionally, we briefly mention the potential geochemical implications of the thermal model to constrain the depth of mantle xenoliths, clearly stating the limitations of the thermal model to constrain deep lithosphere processes.

9. *Ln. 521: the locations of Tihany, Szigliget, Fuzes-to, etc are not shown anywhere in the manuscript.*

   We eliminated this section from the revised ms.